# Shortcut-Resistant CAM Distillation for Long-Tailed Recognition

**Wenhai Wan**[1 2] **Teng Zhang**[1 2] **Shao-Yuan Li**[3 4 5 6] **Xinrui Wang**[3 4 7] **Qiang-Sheng Hua**[1 2] **Songcan Chen**[3 4]

## Abstract

Real-world datasets often follow a long-tailed distribution, making generalization to tail classes difficult. We revisit this problem through the lens of shortcut learning, where models prefer the easiest predictive cues (e.g., background or textures) over object-centric semantics, especially under scarce and biased supervision. We find that this tendency is amplified for tail classes: limited examples often share similar contexts, making non-semantic signals highly correlated and thus tempting shortcuts, whereas head classes with diverse appearances and environments encourage more stable object-focused representations. Motivated by this observation, we propose Shortcut-Resistant CAM Distillation (SRCD), a plug-and-play framework that transfers object-focused explanations from head to tail classes. SRCD operates in the Class Activation Map (CAM) space, where a CAM provides a class-specific spatial evidence map for a prediction. SRCD aggregates CAMs from a small set of head-class candidates into a shortcut-resistant teacher using an energy-model weighting based on coherence and concentration, and distills it to the tail-class CAM. We provide a theoretical analysis that quantifies shortcut reliance as shortcut-region evidence mass in CAM space and shows that SRCD suppresses tail shortcuts. Extensive experiments on long-tailed benchmarks consistently improve strong baselines. The code is available at https://github.com/Haifeng3/SRCD.

[1]National Engineering Research Center for Big Data Technology and System [2]School of Computer Science and Technology, Huazhong University of Science and Technology, Wuhan, China [3]MIIT Key Laboratory of Pattern Analysis and Machine Intelligence [4]College of Computer Science and Technology, Nanjing University of Aeronautics and Astronautics [5]State Key Lab. for Novel Software Technology, Nanjing University [6]the Joint Laboratory of Spatial intelligent Perception and Large Model Application [7]Computer Vision Center, Spain. Correspondence to: Teng Zhang <tengzhang@hust.edu.cn>.

*Proceedings of the 43rd International Conference on Machine Learning*, Seoul, South Korea. PMLR 306, 2026. Copyright 2026 by the author(s).

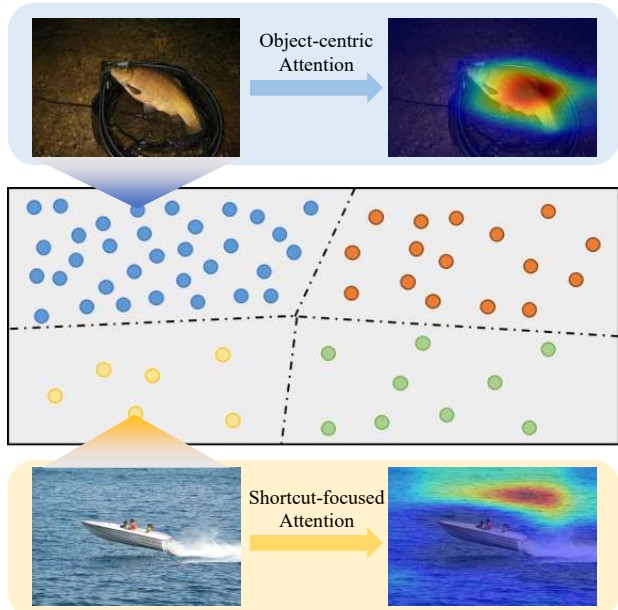

*Figure 1.* Shortcut learning in long-tailed recognition visualized by CAMs. For a head-class example (*Tench*), the model attends to the object region, while for a tail-class example (*Speedboat*), attention drifts to spurious cues (e.g., ocean background), leading to shortcut-dominated predictions and degraded tail generalization.

## 1. Introduction

In recent years, deep neural networks have become the dominant paradigm for visual recognition, enabling strong performance across a wide range of tasks such as image classification (He et al., 2016), medical image analysis (Ronneberger et al., 2015), and object detection (Ren et al., 2015). However, much of this progress implicitly relies on large-scale and relatively balanced training data, which rarely matches the distributions encountered in real-world deployments. In practice, data often follows a long-tailed distribution (Zhang et al., 2023), where a few frequent categories dominate the dataset while most categories contain only a limited number of examples. Learning effectively under such imbalanced supervision has therefore become a central challenge in modern computer vision (Zhang et al., 2025).

Despite steady progress (Zhang et al., 2025), tail-class generalization remains brittle: gains on tail categories are often

modest, and performance can degrade sharply when tail supervision is scarce and biased. Fundamentally, the dominant challenge in long-tailed recognition is still tail data scarcity: with only a few labeled instances, predictors are hard to learn and easy to overfit. Building on this, we make a finer-grained observation: when examples are few, tail data is also more likely to be context-homogeneous (e.g., captured in similar environments), which can make non-semantic cues appear unusually reliable during training. Consequently, even improved optimization or calibration may not prevent the model from settling on undesirable decision rules for tail classes.

A useful lens is to view learning as selecting one explanation among many that fit the training labels equally well (Lee et al., 2023). Under limited supervision, models often favor the easiest explanations, giving rise to shortcut learning (Geirhos et al., 2020; Hermann et al., 2024): instead of discovering object-centric semantics, they may rely on non-semantic yet predictive signals such as background context or characteristic textures. In generic spurious-correlation settings (Ye et al., 2024), such shortcuts are typically attributed to distribution bias in the data. In long-tailed recognition, however, shortcut reliance can be further amplified by scarcity: scarce supervision encourages overfitting to accidental correlations, and the accompanying context homogeneity makes these spurious cues appear *pseudo-stable* within the training set. This scarcity–bias interaction makes shortcuts particularly tempting for minimizing training loss, yet especially harmful for tail generalization.

This mechanism creates a systematic head–tail gap. Head categories contain diverse instances and environments, weakening the reliability of spurious cues and encouraging the model to focus on stable object evidence. In contrast, tail categories often exhibit strong contextual bias across their few examples, making it easier for the model to "explain" them using shortcuts. To make this behavior visible, we use Class Activation Maps (CAMs) (Zhou et al., 2016), which provide a spatial attribution by highlighting image regions that contribute most to a class prediction. As shown in Figure 1, for a head-class example (*Tench*), the CAM concentrates on the object region, indicating meaningful object cues. For a tail-class example (*Speedboat*), scarce and biased supervision makes the ocean background a strong correlate; learning such context is substantially easier than learning the object itself. Consequently, attention drifts toward shortcut cues, leading to shortcut-dominated recognition and systematically poor tail generalization.

Motivated by these observations, we propose Shortcut-Resistant CAM Distillation (SRCD). Different from prior head-to-tail transfer (Li et al., 2024) that propagates knowledge in the feature/logit space, we intervene at the *evidence* level to encourage learning object-centric cues that are essential for tail recognition. Intuitively, since head classes are trained with abundant and diverse examples, their CAMs are more likely to reflect stable object evidence rather than spurious shortcuts. SRCD constructs a shortcut-resistant teacher for each tail sample by aggregating CAMs from a small set of predicted head-class candidates, and distills it to the tail-class CAM. To ensure reliability, SRCD assigns each candidate a lightweight weight based on coherence (Laplacian smoothness) and concentration (Hoyer sparsity), discouraging fragmented or overly diffuse explanations. SRCD is plug-and-play and can be seamlessly combined with existing long-tailed learning methods to further reduce shortcut reliance and improve tail generalization.

Our contributions are three-fold:

- We revisit long-tailed recognition from a shortcut-learning perspective and identify a systematic head–tail gap where tail classes are more prone to shortcut-dominated attention.

- We propose SRCD, a simple and compatible CAM distillation strategy that transfers object-focused explanations from head to tail classes.

- We provide theoretical support and extensive experiments showing that SRCD effectively suppresses shortcut reliance and consistently improves performance on long-tailed benchmarks.

## 2. Related Work

**Long-tailed Recognition.** Long-tailed recognition aims to improve underrepresented tail classes while maintaining head accuracy. To alleviate tail scarcity, a promising direction is head-to-tail transfer that injects diversity into tail categories, e.g., by synthesizing tail-like samples/features (M2m (Kim et al., 2020), RSG (Wang et al., 2021)) or transferring distributional statistics and contexts to enrich tail representations (FASA (Zang et al., 2021), LTA (Yin et al., 2019), CMO (Park et al., 2022), H2T (Li et al., 2024)). Different from these approaches that mainly operate in the sample/feature/statistics space, our method intervenes at the evidence level, encouraging tail predictions to rely on object-centric cues rather than shortcuts. For completeness, we provide more related work and discussion in the appendix A.1.

**Shortcut Learning.** Shortcut learning (Geirhos et al., 2020) describes a common failure mode where deep networks rely on spurious but predictive cues (e.g., background (Beery et al., 2018) context or textures (Geirhos et al., 2018)) instead of semantic evidence. Prior work has studied such shortcuts in standard recognition settings (Hermann et al., 2024) and proposed interventions via augmentation, bias mitigation (Sagawa et al., 2019). However, the

role of shortcut learning under long-tailed distributions remains largely unexplored: tail classes suffer from scarce and biased supervision, making shortcuts unusually consistent and easy to fit, yet existing long-tailed methods rarely address this issue explicitly. We provide more related work in Appendix A.2.

## 3. Methodology

### 3.1. Problem Setup

Let $\mathcal{X}$ be the input space and $\mathcal{Y} = \{1, 2, \ldots, C\}$ the label space. We focus on long-tailed recognition, where the training set $\mathcal{S} = \{(x_i, y_i)\}_{i=1}^n$ is drawn from an imbalanced distribution $\mathcal{D}$, with $x_i \in \mathcal{X}$ and $y_i \in \mathcal{Y}$. The total number of instances is $n$. For each class $y \in \mathcal{Y}$, let $n_y$ denote the number of training examples for that class, and define the empirical class prior as $\pi_y = n_y/n$. We assume that classes are sorted by frequency such that $n_1 \geq n_2 \geq \cdots \geq n_C$ with $n_1 \gg n_C$, capturing the head-to-tail imbalance. Following previous work (Zhang et al., 2021b), we use a frequency threshold $n_{\text{thres}} = 100$ to partition classes into head and tail groups: $\mathcal{Y}_{\text{head}} = \{y \mid n_y \geq n_{\text{thres}}\}$ and $\mathcal{Y}_{\text{tail}} = \{y \mid n_y < n_{\text{thres}}\}$. Note that the above binary partition is used for applying SRCD during training. For split-wise evaluation, we follow the common reporting protocol where Head, Med, and Tail denote classes with more than 100, 20–100, and fewer than 20 training samples, respectively.

Following common long-tailed evaluation protocols (Cao et al., 2019), we consider a balanced test distribution $\mathcal{D}_{bal}$ that assigns equal probability to each class. We further assume that $\mathcal{D}$ and $\mathcal{D}_{bal}$ share the same class-conditional distributions, i.e., $\mathcal{D}(x \mid y) = \mathcal{D}_{bal}(x \mid y)$, for $\forall y \in \mathcal{Y}$, and only differ in their class priors.

Let $f(\cdot; \theta)$ be a classifier parameterized by $\theta \in \mathbb{R}^k$, and let $\ell(\theta; x, y)$ denote the per-example loss. The empirical risk on $\mathcal{S}$ is

$$L_{\mathcal{S}}(\theta) = \frac{1}{n} \sum_{i=1}^n \ell(\theta; x_i, y_i).$$

At the distribution level, the expected risks under $\mathcal{D}$ and $\mathcal{D}_{bal}$ are given by $L_{\mathcal{D}}(\theta) = \mathbb{E}_{(x,y) \sim \mathcal{D}}[\ell(\theta; x, y)]$, and $L_{\mathcal{D}_{bal}}(\theta) = \mathbb{E}_{(x,y) \sim \mathcal{D}_{bal}}[\ell(\theta; x, y)]$, respectively.

Our goal is to learn parameters $\theta$ from the long-tailed training set $\mathcal{S}$ such that the classifier achieves low risk on the balanced test distribution, i.e., minimizing $L_{\mathcal{D}_{bal}}(\theta)$, which reflects strong performance across both head and tail classes.

### 3.2. Revisiting Class Activation Maps

To analyze shortcut learning in long-tailed recognition, we need a mechanism to identify *where* a model focuses its class-specific evidence. Standard metrics fall short, as

models can predict correctly while relying on spurious cues. Since shortcut learning manifests in spatial attention, feature-level interpretability is crucial.

Class Activation Maps (CAMs) (Zhou et al., 2016) provide a principled way to expose such behavior by producing class-specific spatial explanations for a given input. For an image $x$ and class $c$, CAM highlights regions that contribute most to the prediction, making it a suitable tool for diagnosing whether the model attends to meaningful object evidence or shortcut cues.

Formally, let $\{A_k(x)\}_{k=1}^K$ denote the feature maps from a convolutional layer. The class activation map for class $c$ is defined as

$$M_c(x) = \phi\left(\sum_{k=1}^K \alpha_k^c(x) A_k(x)\right) \in \mathbb{R}^{H \times W},$$

where $\alpha_k^c(x)$ measures the contribution of the $k$-th feature map to class $c$, and $\phi(\cdot)$ is a non-linear operation (e.g., ReLU). In practice, $\alpha_k^c(x)$ can be obtained using standard CAM (Zhou et al., 2016), which require no architectural modifications and generalize well across models.

To facilitate quantitative analysis, we interpret CAMs as spatial evidence distributions. Denoting by $M_c(x)_u$ the activation at spatial location $u \in \{1, \ldots, HW\}$, we normalize the CAM as

$$\bar{M}_c(x) = \frac{M_c(x)}{\sum_u M_c(x)_u + \varepsilon}, \tag{1}$$

The normalized map $\bar{M}_c(x)$ represents the fraction of class-specific evidence allocated to each location. This formulation enables direct comparison of attention patterns across classes and examples, and serves as the foundation for our shortcut diagnosis and distillation framework.

### 3.3. Diagnosing Shortcut Learning in Long-Tailed Recognition

To empirically diagnose shortcut learning in long-tailed recognition, we leverage CAMs as a spatial indicator of what evidence a model relies on. Our key hypothesis is that models trained on a balanced dataset are more likely to learn object-centric semantics, since abundant and diverse supervision reduces the chance of overfitting to spurious correlations. Therefore, CAMs produced by a balanced model can serve as a reasonable reference for semantic explanations, while deviations from this reference may reflect shortcut-dominated attention. A more detailed discussion on why CAM alignment with the balanced reference is a reasonable proxy for shortcut reliance (together with limitations) is deferred to Appendix B.1.

Concretely, we train a reference model on the full balanced CIFAR100 (Krizhevsky et al., 2009) and another model on

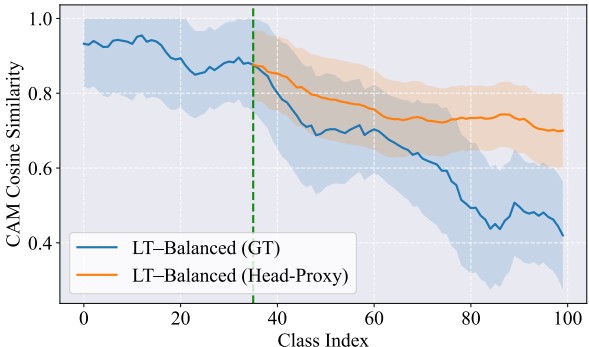

*Figure 2.* CAM similarity analysis. **LT–Balanced (GT)** denotes the cosine similarity between CAMs produced by the long-tailed model and the balanced model for the same examples under their ground-truth classes. **LT–Balanced (Head-Proxy)** computes CAMs for tail examples using the most probable head-class predictions from the long-tailed model and compares them to the balanced-model CAMs. Shaded regions indicate standard deviation.

its long-tailed counterpart CIFAR100-LT (Cao et al., 2019). For the same input example $x$ with label $y$, we compute the class-specific CAMs from both models, denoted as $M_y^{bal}(x)$ and $M_y^{lt}(x)$, respectively, and measure their alignment via cosine similarity:

$$\mathrm{Sim}(x,y) = \cos\big(\mathrm{vec}(\bar{M}_y^{lt}(x)),\ \mathrm{vec}(\bar{M}_y^{bal}(x))\big), \quad (2)$$

where $\bar{M}$ indicates normalization as in Eq. (1), and $\mathrm{vec}(\cdot)$ vectorizes the spatial map. Intuitively, a high similarity suggests that the long-tailed model attends to regions consistent with the balanced reference (i.e., less shortcut reliance), whereas a low similarity indicates that the long-tailed model may exploit alternative, non-semantic cues to explain the same class prediction.

Figure 2 reports the CAM cosine similarity across classes, sorted from head to tail according to training frequency. We observe that for head classes, the long-tailed model produces CAMs highly consistent with the balanced reference, indicating that shortcut reliance is relatively mild when sufficient supervision is available. In contrast, the similarity drops substantially on tail classes, revealing a systematic shift in spatial evidence under scarce training data. This supports the intuition that limited tail supervision makes spurious correlations easier to fit, leading to shortcut-dominated explanations and degraded tail generalization.

Moreover, we examine whether tail examples can benefit from object-centric explanations learned by head classes. For each tail example $x$, we select the most confident predicted head class $\hat{k}$ and compute its CAM $M_{\hat{k}}^{lt}(x)$. We then compare this *head-proxy* CAM to the balanced reference CAM $M_y^{bal}(x)$ of the ground-truth class. As shown

in Figure 2, the head-proxy CAM exhibits notably higher similarity on tail classes than the tail-class CAM itself. This suggests that borrowing CAM cues from a likely head class can provide a more stable, less shortcut-prone spatial explanation for tail examples, motivating our shortcut-resistant CAM distillation framework.

### 3.4. Shortcut-Resistant CAM Distillation

In this section, we present **Shortcut-Resistant CAM Distillation (SRCD)**, a training strategy that explicitly suppresses shortcut-dominated explanations for tail classes by distilling reliable spatial evidence from head classes. SRCD is motivated by our empirical findings that (i) tail-class CAMs learned under long-tailed supervision deviate substantially from those obtained under balanced training, and (ii) CAMs computed using predicted head classes provide more stable and semantically aligned explanations for tail examples.

**Motivation.** Under long-tailed distributions, tail classes suffer from limited and biased supervision, making it easy for models to rely on spurious but predictive cues to explain these classes. Such shortcut reliance manifests as fragmented or overly diffuse spatial attention. In contrast, head classes are trained with abundant and diverse examples, which reduces the consistency of spurious correlations and encourages the model to learn more robust visual evidence. SRCD leverages this asymmetry by constructing a shortcut-resistant teacher explanation from head-class CAMs and transferring it to tail classes through spatial distillation.

**Head-Class Proxy Selection.** Given a training example $(x,y)$ with $y \in \mathcal{Y}_{\text{tail}}$, we first compute the predictive distribution $p(c \mid x)$. From the head-class set $\mathcal{Y}_{\text{head}}$, we select the top-$K$ most probable classes:

$$\mathcal{H}(x) = \arg \max_{c \in \mathcal{Y}_{\text{head}}}^{K} p(c \mid x).$$

Each $c \in \mathcal{H}(x)$ defines a *head-class proxy* for $x$, and induces a candidate explanation $\bar{M}_c(x)$. As shown in Section 3.3, such head-proxy CAMs are empirically less shortcut-prone than tail-class CAMs computed using the ground-truth label. This suggests that head proxies can provide more stable spatial evidence for tail examples, even without enforcing strict semantic matching or explicit part-level decomposition. We provide further discussion on this design in Appendix B.3.

**Quantifying Shortcut Resistance of CAMs.** Not all head-proxy CAMs are equally reliable. Some may still be noisy, fragmented, or overly diffuse. To identify explanations that are resistant to shortcut reliance, we evaluate each candidate CAM using two complementary structural properties.

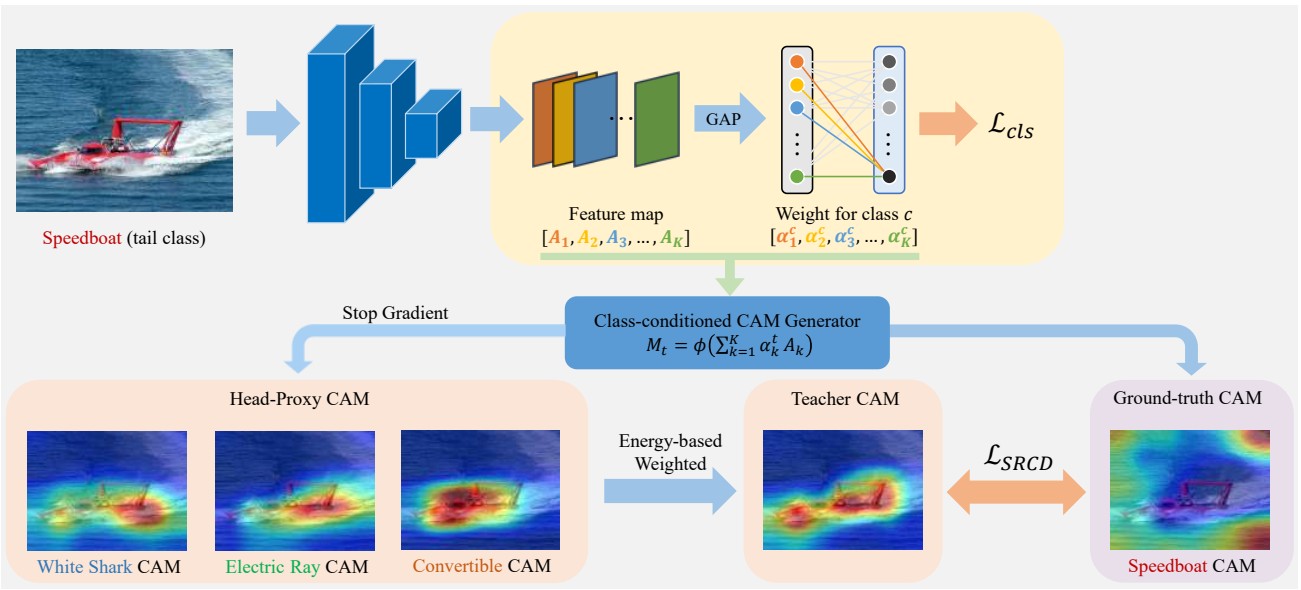

*Figure 3.* SRCD pipeline. We generate class-conditioned CAMs from feature maps and classifier weights, form a shortcut-resistant teacher CAM by energy-weighting a set of head-proxy CAMs (with stop-gradient used only in this proxy branch), and distill the teacher to the ground-truth class CAM via $\mathcal{L}_{\text{SRCD}}$ to reduce shortcut/background attention alongside $\mathcal{L}_{\text{cls}}$.

*(1) Spatial Coherence Energy.* Shortcut-driven explanations often manifest as scattered activations over multiple disconnected regions, reflecting reliance on incidental cues rather than consistent visual evidence. To penalize such behavior, we measure the spatial coherence of a CAM using a Laplacian-based energy: $\mathcal{E}_{\text{h}}(M) = M^{\top} L M$, where $M \in \mathbb{R}^{HW}$ is the vectorized CAM and $L$ is the graph Laplacian defined on the $H \times W$ grid. Specifically, $L = D - G$, where $G$ encodes 4-neighbor pixel adjacency and $D$ is the corresponding degree matrix. Lower coherence energy indicates smoother and more contiguous spatial evidence, which is less characteristic of shortcut-driven attention.

*(2) Evidence Concentration Score.* Shortcut learning may also result in evidence being diffusely spread across large regions, particularly when background cues dominate the prediction. To discourage such behavior, we measure how concentrated the evidence is using a normalized sparsity score:

$$\mathcal{E}_{\text{n}}(M) = \frac{\sqrt{N} - \|M\|_1 / \|M\|_2}{\sqrt{N} - 1},$$

where $N = HW$, $\|M\|_1$ and $\|M\|_2$ are the $L_1$ and $L_2$ norms of the vectorized CAM. For nonnegative CAMs, $\|M\|_1 / \|M\|_2$ increases when evidence spreads over many locations and decreases when evidence concentrates, making $\mathcal{E}_{\text{n}}(M) \in [0, 1]$ a normalized concentration measure.

Together, $\mathcal{E}_{\text{h}}$ and $\mathcal{E}_{\text{n}}$ capture complementary aspects of shortcut resistance: spatial continuity and evidence compactness. For a more detailed discussion on the rationale and limitations of $\mathcal{E}_h$ and $\mathcal{E}_n$, see Appendix B.2.

**Energy-Based Teacher Aggregation.** We aggregate head-proxy CAMs into a single shortcut-resistant teacher by assigning each candidate an energy-based weight:

$$w_c(x) = \frac{\exp\left(\frac{-\mathcal{E}_{\text{h}}(\bar{M}_c(x)) + \mathcal{E}_{\text{n}}(\bar{M}_c(x))}{\tau}\right)}{\sum_{c' \in \mathcal{H}(x)} \exp\left(\frac{-\mathcal{E}_{\text{h}}(\bar{M}_{c'}(x)) + \mathcal{E}_{\text{n}}(\bar{M}_{c'}(x))}{\tau}\right)}.$$

This formulation favors CAMs that are both spatially coherent and concentrated, while suppressing candidates that exhibit fragmented or diffuse patterns.

The resulting teacher CAM is defined as

$$\bar{M}_{\text{T}}(x) = \sum_{c \in \mathcal{H}(x)} w_c(x)\, \bar{M}_c(x).$$

**Adaptive Distillation to Tail Classes.** Let $\bar{M}_y(x)$ denote the normalized CAM for the tail class $y$ and $\bar{M}_{\text{T}}(x)$ the shortcut-resistant teacher CAM. We align tail explanations with the teacher using an $L_2$ distillation loss weighted by a class-dependent factor:

$$\mathcal{L}_{\text{SRCD}}(x, y) = \gamma_y \big\| \bar{M}_y(x) - \text{sg}\big[\bar{M}_{\text{T}}(x)\big] \big\|_2^2, \quad (3)$$

where $\text{sg}[\cdot]$ denotes the stop-gradient operation, and $\gamma_y$ is a class-dependent coefficient that increases as class $y$ becomes rarer, reflecting the fact that tail classes are more prone to shortcut learning and thus require stronger regularization. In our implementation, we set $\gamma_y = n_{\text{thres}} / n_y$, with $n_y$ the number of training examples for class $y$ and $n_{\text{thres}}$ the

head–tail threshold. This choice reflects the intuition that tail classes with fewer examples are more prone to shortcut learning and therefore require stronger regularization.

**Overall Objective.** We optimize

$$\mathcal{L}(x, y) = \mathcal{L}_{\text{cls}}(x, y) + \mathbb{I}[y \in \mathcal{Y}_{\text{tail}}] \, \beta \, \mathcal{L}_{\text{SRCD}}(x, y),$$

where $\mathcal{L}_{\text{cls}}$ can be any long-tailed re-balancing loss (e.g., LDAM (Cao et al., 2019)) and $\beta$ controls the distillation strength. SRCD is architecture-agnostic and can be plugged into existing long-tailed methods. Since SRCD relies on reliable head-class explanations, we enable it only in the late training stage, which also keeps the overall training lightweight. Additional schedule ablations are deferred to Appendix D.6.

## 4. Theoretical Analysis

We provide a shortcut-centric analysis of SRCD in the CAM space. Our key idea is to measure shortcut learning as the attention mass assigned to shortcut/background regions, and show that SRCD reduces it in a principled way: (i) an energy-weighted teacher is constructed to favor coherent and concentrated (thus object-centric) evidence, and (ii) $L_2$ CAM distillation induces a contraction dynamics that monotonically transports shortcut-region attention mass from tail CAMs toward this teacher. Together, these results explain why SRCD is effective, while we defer detailed proofs and extended discussions to the appendix C.

Let $\bar{M}_y(x) \in \mathbb{R}^{H \times W}$ denote the normalized CAM for class $y$, where $\bar{M}_y(x) \geq 0$ and $\sum_{u \in \Omega} \bar{M}_y(x)_u = 1$. We formalize shortcut reliance as the CAM mass on a shortcut/background region $\Omega_{\text{sc}} \subset \Omega$:

$$\mathcal{A}_{\text{sc}}(\bar{M}) = \sum_{u \in \Omega_{\text{sc}}} \bar{M}_u \in [0, 1]. \tag{4}$$

**Energy-weighted teacher as a principled utility maximizer.** SRCD constructs a shortcut-resistant teacher CAM by aggregating head-proxy candidates $\{\bar{M}_c(x)\}_{c \in \mathcal{H}(x)}$ with the following weights:

$$\bar{M}_T(x) = \sum_{c \in \mathcal{H}(x)} w_c(x) \, \bar{M}_c(x),$$

$$w_c(x) = \frac{\exp\big(\mathcal{U}(\bar{M}_c(x))\big)}{\sum_{c' \in \mathcal{H}(x)} \exp\big(\mathcal{U}(\bar{M}_{c'}(x))\big)}, \tag{5}$$

where the utility is defined by

$$\mathcal{U}(\bar{M}) = -\mathcal{E}_h(\bar{M}) + \mathcal{E}_n(\bar{M}),$$

with $\mathcal{E}_h$ encouraging spatial coherence and $\mathcal{E}_n$ encouraging evidence concentration.

**Theorem 4.1** (Entropy-regularized utility maximization). *Let $\mathcal{U}_c = \mathcal{U}(\bar{M}_c(x))$ for $c \in \mathcal{H}(x)$. The softmax weights in (5) are the unique maximizer of the strictly concave objective*

$$\max_{w \in \Delta} \sum_c w_c \mathcal{U}_c - \tau \sum_c w_c \log w_c, \tag{6}$$

*where $\Delta = \{w : w_c \geq 0, \sum_c w_c = 1\}$ and $\tau > 0$ controls the smoothness of aggregation.*

Eq. (6) shows that teacher construction is principled: it prefers CAMs with low coherence energy $\mathcal{E}_h$ (more spatially consistent evidence) and high concentration score $\mathcal{E}_n$ (more object-centric evidence), while the entropy term prevents overly brittle selection.

**$L_2$ CAM distillation induces shortcut-mass transport.** SRCD distills the teacher into the tail-class CAM using Eq (3) where $\gamma_y$ increases with tail scarcity.

**Theorem 4.2** (Monotone shortcut-mass contraction). *Consider the gradient flow that minimizes the $L_2$ distillation energy:*

$$\frac{d}{dt} \bar{M}(t) = -\nabla_{\bar{M}} \big\| \bar{M}(t) - \bar{M}_T \big\|_2^2. \tag{7}$$

*Then $\bar{M}(t)$ admits a closed-form solution $\bar{M}(t) = \bar{M}_T + e^{-2t}\big(\bar{M}(0) - \bar{M}_T\big)$, and*

$$\mathcal{A}_{\text{sc}}(\bar{M}(t)) = \mathcal{A}_{\text{sc}}(\bar{M}_T) + e^{-2t}\Big(\mathcal{A}_{\text{sc}}(\bar{M}(0)) - \mathcal{A}_{\text{sc}}(\bar{M}_T)\Big). \tag{8}$$

*In particular, if $\mathcal{A}_{\text{sc}}(\bar{M}(0)) > \mathcal{A}_{\text{sc}}(\bar{M}_T)$, then $\mathcal{A}_{\text{sc}}(\bar{M}(t))$ decreases monotonically and converges exponentially to $\mathcal{A}_{\text{sc}}(\bar{M}_T)$.*

Eq. (8) shows that $L_2$ distillation makes $\bar{M}(t)$ exponentially contract to $\bar{M}_T$. Since $\mathcal{A}_{\text{sc}}(\cdot)$ is a linear mass on $\Omega_{\text{sc}}$, the shortcut attention is transported toward $\mathcal{A}_{\text{sc}}(\bar{M}_T)$ at the same rate; thus if the teacher is more object-centric, shortcut reliance decreases monotonically.

**SRCD suppresses tail shortcuts.** If the energy-weighted teacher is shortcut-resistant (i.e., $\mathcal{A}_{\text{sc}}(\bar{M}_T)$ is small), then minimizing Eq (3) drives the tail CAM $\bar{M}_y(x)$ toward $\bar{M}_T(x)$ and consequently reduces shortcut reliance $\mathcal{A}_{\text{sc}}(\bar{M}_y(x))$ toward $\mathcal{A}_{\text{sc}}(\bar{M}_T(x))$.

## 5. Experiments

### 5.1. Experiment Protocols

**Datasets.** We evaluate our method on five widely adopted long-tailed recognition benchmarks: CIFAR-10-LT and

*Table 1.* Training-from-scratch results on CIFAR-10-LT and CIFAR-100-LT under imbalance ratios IR $\in \{100, 50, 10\}$ (Top-1 accuracy, %). +SRCD denotes integrating SRCD into each baseline; superscripts show absolute changes (↑/↓ for gain/drop).

| Method | CIFAR100 | | | CIFAR10 | | |
|---|---|---|---|---|---|---|
| | IR=100 | IR=50 | IR=10 | IR=100 | IR=50 | IR=10 |
| BCL (Zhu et al., 2022) | 51.76 | 56.51 | 67.90 | 83.61 | 86.13 | 90.10 |
| +SRCD | 53.64↑1.88 | 56.81↑0.30 | 68.22↑0.32 | 82.97↓0.64 | 87.87↑1.74 | 91.23↑1.13 |
| GCL (Li et al., 2022b) | 46.50 | 51.72 | 61.79 | 80.56 | 84.74 | 89.65 |
| +SRCD | 47.07↑0.57 | 52.32↑0.60 | 60.23↓1.56 | 81.11↑0.55 | 85.69↑0.95 | 88.90↓0.75 |
| CE+FCC (Li et al., 2023) | 40.20 | 45.93 | 57.80 | 73.80 | 79.57 | 87.75 |
| +SRCD | 42.38↑2.18 | 47.17↑1.24 | 58.29↑0.49 | 75.29↑1.49 | 80.46↑0.89 | 88.05↑0.30 |
| GLMC (Du et al., 2023) | 53.91 | 58.87 | 68.07 | 83.68 | 86.90 | 91.16 |
| +SRCD | 55.26↑1.35 | 60.43↑1.56 | 69.01↑0.94 | 83.97↑0.29 | 87.72↑0.82 | 90.78↓0.38 |
| CE+ H2T (Li et al., 2024) | 42.27 | 47.58 | 58.24 | 79.91 | 82.80 | 88.77 |
| +SRCD | 44.53↑2.26 | 46.97↓0.61 | 60.07↑1.83 | 80.43↑0.52 | 83.29↑0.49 | 89.21↑0.44 |
| BNS (Lin & Yuan, 2025) | 52.40 | 55.90 | 62.60 | 76.40 | 81.90 | 83.70 |
| +SRCD | 53.88↑1.48 | 56.12↑0.22 | 63.04↑0.44 | 77.36↑0.96 | 81.66↓0.24 | 84.85↑1.15 |
| Focal-SAM (Li et al., 2025) | 50.70 | 54.50 | 63.80 | 82.90 | 85.50 | 90.50 |
| +SRCD | 52.54↑1.84 | 55.29↑0.79 | 64.71↑0.91 | 84.29↑1.39 | 86.04↑0.54 | 90.77↑0.27 |

CIFAR-100-LT (Cao et al., 2019), ImageNet-LT (Liu et al., 2019), iNaturalist 2018 (Van Horn et al., 2018), and Places-LT (Liu et al., 2019). We defer details to Appendix D.1.

**Evaluation Settings and Competitors.** We evaluate our method under two practical settings: *training from scratch* and *fine-tuning foundation models*. For training from scratch, we compare against representative long-tailed learning baselines, including BCL (Zhu et al., 2022), GCL (Li et al., 2022b), FCC (Li et al., 2023), GLMC (Du et al., 2023), H2T (Li et al., 2024), BNS (Lin & Yuan, 2025), and Focal-SAM (Li et al., 2025). For fine-tuning foundation models, we follow the standard CLIP adaptation protocol and compare with recent methods such as Decoder (Wang et al., 2024b), PECEL (Ru et al., 2024) and LIFT (Shi et al., 2024). Notably, our method is a plug-and-play component that can be seamlessly combined with arbitrary long-tailed learning strategies across both settings.

**Implementation Details.** We report results under two settings: *training from scratch* and *fine-tuning foundation models*. For the scratch setting, we adopt ResNet-32 (He et al., 2016) on CIFAR-LT and train for 200 epochs, while using ResNet-50 (He et al., 2016) on ImageNet-LT (Liu et al., 2019) and iNaturalist (Van Horn et al., 2018) with the same training schedule. For foundation model fine-tuning, we follow the protocol in LIFT (Shi et al., 2024) and fine-tune the image encoder of CLIP (Radford et al., 2021) with a ViT-B/16 (Dosovitskiy, 2020) backbone for 20 epochs. Additional training details are provided in the Appendix.

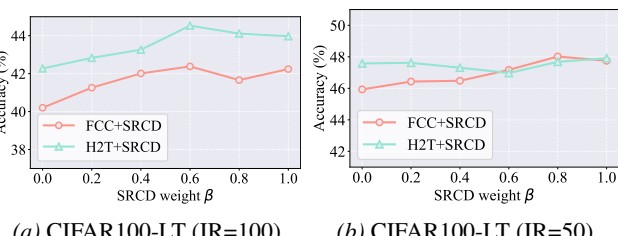

*(a)* CIFAR100-LT (IR=100)  *(b)* CIFAR100-LT (IR=50)

*Figure 4.* Ablation study of SRCD w.r.t. $\beta$.

## 5.2. Results with training from scratch

We report *training from scratch* results on CIFAR10/100-LT (Table 1) and large-scale long-tailed benchmarks (upper block of Table 2), where SRCD is integrated into a wide range of representative baselines. Overall, SRCD delivers consistent gains across imbalance ratios and datasets, indicating that explanation-level shortcut suppression is complementary to existing long-tailed objectives rather than method-specific. On CIFAR100-LT, adding SRCD improves most baselines under IR $\in \{100, 50, 10\}$, with larger gains typically observed in the more challenging regimes, while CIFAR10-LT shows similarly stable improvements across skews. This trend scales to ImageNet-LT and iNaturalist: SRCD consistently increases the overall accuracy for all evaluated methods and yields particularly strong improvements on Medium/Tail splits, which aligns with our motivation that data-scarce classes are more prone to relying on shortcut/background cues. Additional split-wise results are provided in Appendix D.3, offering a more detailed analysis of SRCD's effect on tail-class performance.

*Table 2.* Results on ImageNet-LT and iNaturalist, reporting Top-1 accuracy (%) on Head/Med/Tail splits and All. The upper block is training from scratch, and the lower block is fine-tuning foundation models. +SRCD denotes integrating SRCD into each baseline; superscripts show absolute changes (↑/↓ for gain/drop).

| Method | ImageNet-LT | | | | iNaturalist | | | |
|---|---|---|---|---|---|---|---|---|
| | Head | Med | Tail | All | Head | Med | Tail | All |
| **Training from scratch** | | | | | | | | |
| BCL | 65.83 | 53.16 | 36.27 | 55.67 | 73.47 | 71.99 | 68.82 | 70.80 |
| +SRCD | 66.29↑0.46 | 54.47↑1.31 | 39.56↑3.29 | 57.00↑1.33 | 74.26↑0.79 | 72.65↑0.66 | 70.33↑1.51 | 71.89↑1.09 |
| GCL | 65.04 | 52.77 | 35.69 | 55.51 | 73.75 | 72.28 | 68.34 | 70.76 |
| +SRCD | 64.83↓0.21 | 53.62↑0.85 | 37.10↑1.41 | 55.69↑0.18 | 74.28↑0.53 | 74.02↑1.74 | 70.31↑1.97 | 72.58↑1.82 |
| CE+FCC | 65.52 | 52.18 | 33.83 | 54.68 | 72.78 | 71.09 | 67.41 | 69.70 |
| +SRCD | 66.15↑0.63 | 53.29↑1.11 | 36.48↑2.65 | 55.95↑1.27 | 72.79↑0.01 | 70.84↓0.25 | 68.87↑1.46 | 70.26↑0.56 |
| GLMC | 69.42 | 52.49 | 30.41 | 56.17 | 76.04 | 73.69 | 71.77 | 73.14 |
| +SRCD | 70.43↑1.01 | 52.17↓0.32 | 34.22↑3.81 | 56.76↑0.59 | 76.77↑0.73 | 73.21↓0.48 | 73.06↑1.29 | 73.52↑0.38 |
| Focal-SAM | 63.90 | 52.20 | 34.40 | 54.30 | 68.40 | 72.00 | 72.50 | 71.80 |
| +SRCD | 63.99↑0.09 | 52.71↑0.51 | 36.67↑2.27 | 54.87↑0.57 | 69.03↑0.63 | 73.14↑1.14 | 72.43↓0.07 | 72.43↑0.63 |
| **Fine-tuning foundation model** | | | | | | | | |
| Decoder | 76.0 | 72.1 | 69.3 | 73.2 | 55.1 | 58.6 | 61.1 | 59.2 |
| +SRCD | 76.3↑0.3 | 73.4↑1.3 | 71.2↑1.9 | 74.2↑1.0 | 58.7↑3.6 | 61.7↑3.1 | 61.4↑0.3 | 61.3↑2.1 |
| PECEL | 81.3 | 77.6 | 73.0 | 78.3 | 76.8 | 80.6 | 81.3 | 80.5 |
| +SRCD | 81.8↑0.5 | 77.5↓0.1 | 74.6↑1.6 | 78.8↑0.5 | 77.4↑0.6 | 81.2↑0.6 | 82.3↑1.0 | 81.2↑0.7 |
| LIFT | 79.7 | 76.2 | 72.8 | 77.1 | 74.1 | 79.4 | 81.5 | 79.7 |
| +SRCD | 80.4↑0.7 | 77.7↑1.5 | 75.2↑2.4 | 78.4↑1.3 | 75.2↑1.1 | 80.9↑1.5 | 80.8↓0.7 | 80.3↑0.6 |

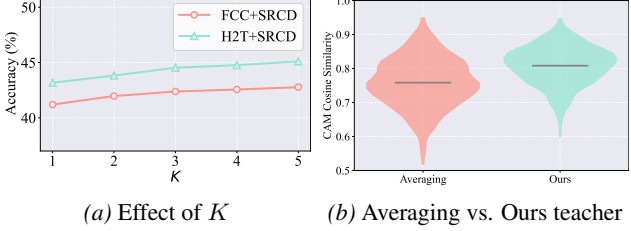

*(a)* Effect of $K$     *(b)* Averaging vs. Ours teacher

*Figure 5.* Ablation study of $K$ and energy-based weighting.

*Table 3.* Ablation study on the teacher-weighting cues $\mathcal{E}_h$ and $\mathcal{E}_n$ in SRCD, evaluated on ImageNet-LT with the FCC.

| Coherence $\mathcal{E}_h$ | Concentration $\mathcal{E}_n$ | Acc. |
|---|---|---|
| ✓ | ✗ | 54.02 |
| ✗ | ✓ | 53.64 |
| ✓ | ✓ | **55.95** |

and LIFT), and the gains are achieved alongside stable or improved Head performance, suggesting that SRCD enhances non-head generalization without trading off head-class accuracy. These results support SRCD as a lightweight, plug-and-play regularizer that remains effective when adapting foundation models to long-tailed distributions.

## 5.3. Results with fine-tuning foundation models

We further evaluate SRCD under the *foundation model fine-tuning* setting (lower block of Table 2), where we plug SRCD into strong long-tailed fine-tuning baselines without modifying architectures. Additional Places-LT results are provided in Appendix D.3, further demonstrating the generality of SRCD on large-scale long-tailed benchmarks. Across all evaluated methods, SRCD consistently improves the overall accuracy on both ImageNet-LT and iNaturalist, while delivering clear gains on Medium/Tail splits, which are typically the most sensitive to shortcut reliance. In particular, SRCD provides substantial tail improvements when paired with competitive fine-tuning pipelines (e.g., PECEL

## 5.4. Ablation Study

**Impact of $\beta$.** We vary $\beta$ to control the distillation strength (Figure 4). Notably, when $\beta$ exceeds 0.6, we observe a significant and stable improvement in accuracy, indicating that stronger distillation has a regularizing effect on tail-class attention.

**Impact of $K$.** We ablate the number of head-class proxies $K$ used to build the teacher (Figure 5 (a)). Using $K = 3$ already provides most of the gains, and further increasing

$K$ leads to stable but marginal improvements.

**Impact of $\mathcal{E}_h$ and $\mathcal{E}_n$.** We toggle the coherence cue $\mathcal{E}_h$ and concentration cue $\mathcal{E}_n$ in teacher weighting (Table 3). Using both cues achieves the best performance.

**Comparison with Averaging Strategy.** We compare our energy-based aggregation with simple averaging for teacher CAM construction (Figure 5 (b)) using Eq (2). Energy-based weighting yields higher CAM similarity to the balanced reference model, indicating that the resulting teacher is more shortcut-resistant.

**Comparison with Center Mask Prior.** We compare SRCD with a center-mask baseline to test whether its gain comes from a trivial geometric prior. Replacing the head-proxy teacher with a fixed center mask leads to degraded performance, especially on the Tail split, while SRCD consistently improves the baseline. This shows that SRCD benefits from adaptive shortcut-resistant evidence transfer rather than simple center-biased attention. Full results are provided in Table 9.

**Additional Analyses.** We further report the computational overhead and provide qualitative CAM visualizations to better illustrate SRCD's behavior, and these results are deferred to the Appendix.

## 6. Conclusion

We introduce SRCD, a method to reduce shortcut learning in long-tailed recognition. By distilling reliable spatial evidence from head classes, SRCD aligns tail-class explanations with more stable object features. Our experiments show that SRCD improves tail-class performance while maintaining accuracy on head classes, offering a robust regularizer for long-tailed learning tasks.

## Acknowledgements

This work was supported by the National Science and Technology Major Project (2024YFB3311401), the National Natural Science Foundation of China (62576168), Open Project Funds for the Joint Laboratory of Spatial Intelligent Perception and Large Model Application (SIPLMA-2024-YB-05), the Natural Science Foundation of Wuhan (2023010201020229), and the Fundamental Research Funds for the Central Universities (NO.NJ2023032).

## Impact Statement

This paper presents work whose goal is to advance the field of Machine Learning. There are many potential societal consequences of our work, none which we feel must be specifically highlighted here.

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

# A. Related Works

## A.1. Long-tailed Recognition and Head-to-Tail Transfer

Long-tailed recognition aims to improve the performance of underrepresented tail classes while maintaining accuracy on head classes. Early solutions mainly resorted to resampling (Chawla et al., 2002; Wang et al., 2020; Zang et al., 2021) or re-weighting (Luo et al., 2025; Park et al., 2021) to rebalance the class distribution during training, e.g., by increasing the effective contribution of tail instances or reducing the dominance of head classes in gradient updates. While such re-balancing is simple and often effective, it may introduce optimization instability (e.g., repeated exposure to a small set of tail samples) and can potentially degrade head performance when the re-balancing is aggressive.

Data augmentation provides another route to alleviate tail scarcity by expanding the diversity of rare categories. A line of work transfers semantic knowledge from head classes to enrich tail representations (Kim et al., 2020; Wang et al., 2021), whereas other methods synthesize new examples or features based on prior information and model feedback (Zang et al., 2021; Li et al., 2021). These approaches generally aim to compensate for the missing intra-class variation of tail classes, either by generating tail-like samples/features or by expanding the tail distribution in a controlled manner.

To improve prediction calibration under imbalance, logit adjustment methods shift the decision boundary according to class frequency, encouraging more balanced margins and reducing the bias toward head classes (Li et al., 2022a; Menon et al., 2021; Zhang et al., 2021a). Since many re-balancing strategies can trade head performance for tail gains, recent advances have further explored decoupled training pipelines that separate representation learning from classifier re-balancing (Kang et al., 2019; Lin & Yuan, 2025; Sun et al., 2025). In parallel, contrastive learning variants have been adopted to learn more robust and separable representations under severe imbalance by reshaping instance- and class-level similarity structures (Cui et al., 2021; Zhu et al., 2022; Hou et al., 2023; Xuan & Zhang, 2024).

**Head-to-tail transfer for alleviating tail scarcity.** A promising direction to address data scarcity in long-tailed learning is to transfer diverse information from head classes to augment tail representations. M2m (Kim et al., 2020) pioneers a generative major-to-minor translation framework based on adversarial learning. It learns a generator that maps abundant majority samples into the feature space of minority classes, effectively upsampling tail data with high-fidelity synthetic counterparts and enabling training on a more balanced set without collecting new real-world samples.

Moving beyond raw image generation, RSG (Wang et al., 2021) augments tail classes at the feature level via rare style generation. It designs a learned displacement field to extract intra-class variations from head classes and applies them to tail prototypes, thereby synthesizing diverse tail features by perturbing tail centers with rich head-induced variations.

Focusing on feature statistics, FASA (Zang et al., 2021) combines feature augmentation with adaptive resampling. It models each class as a Gaussian cloud and leverages statistics transferred from head data to estimate the missing variance of tail classes. Augmented features are then sampled dynamically (often guided by training status or model performance) to maintain balanced optimization throughout training.

LTA (Yin et al., 2019) further emphasizes second-order statistics, particularly covariance transfer from head to tail. By assuming that semantically related classes share similar directions of intra-class variation, LTA inflates point-like tail representations into fuller distributions, allowing the classifier to anticipate unseen variations and smoothing decision boundaries for rare categories.

In the realm of data mixing, CMO (Park et al., 2022) proposes a CutMix-based context-rich minority oversampling strategy. It samples diverse background contexts from head images and pastes them onto tail objects to form composite training examples, transferring environmental diversity to tail classes and reducing overfitting to limited tail backgrounds.

Most recently, H2T (Li et al., 2024) develops a head-to-tail feature fusion mechanism to improve the transferability of semantics. It identifies transferable semantic statistics (e.g., prototypes and variance) from head classes and fuses them with tail features to supplement missing diversity, effectively enriching tail representations without relying on heavy generative models.

## A.2. Shortcut Learning and Spurious Correlations

**Definition and mechanisms.** Deep neural networks, despite strong in-distribution performance, often exhibit *shortcut learning* (Geirhos et al., 2020), i.e., relying on superficial yet predictive cues that fit the training distribution but fail under distribution shift. This behavior is closely related to the *simplicity bias* of networks trained with SGD: models tend to

adopt low-complexity features that minimize the training loss (e.g., color, texture, background) rather than more semantic but harder-to-learn evidence. As a result, standard empirical risk minimization can produce models that are correct for non-robust reasons, leading to poor OOD generalization.

**Manifestations in vision.** In visual recognition, shortcut learning commonly manifests as context and texture biases. Background/context bias arises when models spuriously correlate the environment with the label; for example, models trained on "cows in pastures" may fail on "cows on beaches" (Beery et al., 2018), indicating an over-reliance on background signals instead of object evidence. Related analyses further quantify this dependency by measuring performance drops when foreground and background statistics are altered or separated (Xiao et al., 2020). Texture bias reflects another prevalent shortcut: convolutional networks tend to prefer local texture cues over global shape information (Geirhos et al., 2018), making predictions brittle under style transfer and domain shift.

**Mitigation strategies.** A broad set of methods has been proposed to mitigate spurious correlations and improve robustness. Invariant learning aims to extract features that remain stable across environments; Invariant Risk Minimization (IRM) enforces that the same classifier is optimal across multiple training environments, thereby discouraging reliance on environment-specific spurious cues (Arjovsky et al., 2019). Robust optimization approaches, such as Distributionally Robust Optimization (DRO), improve worst-group performance by explicitly optimizing for challenging sub-populations where shortcuts break (Sagawa et al., 2019). Causal-inspired interventions attempt to break the dependence between shortcuts and labels by manipulating the data generation process; counterfactual augmentation removes or alters spurious attributes (Agarwal et al., 2020), while causal attention mechanisms encourage models to attend to causal evidence instead of contextual confounders.

Beyond conventional shortcut learning, recent studies have also discussed myopic learning behaviors in other settings. For example, online continual learning may suffer from model myopia, where models rely on locally convenient features rather than learning globally robust representations (Wang et al., 2024a). Similar concerns have been raised in positive-unlabeled learning, where holistic predictive trends are used to mitigate decisions based on limited or isolated supervision signals (Xinrui et al., 2023). Although these settings differ from long-tailed recognition, they share a common intuition: limited or locally biased supervision can make easy but suboptimal cues overly attractive. Our work studies this issue from an evidence-level perspective, showing that tail-class scarcity can induce shortcut-dominated CAMs and that SRCD suppresses such spatial shortcut reliance.

## B. More Discussion

### B.1. Additional Discussion on CAM-Based Shortcut Diagnosis

Here, we provide additional rationale for using CAM similarity to a balanced reference model as a proxy for shortcut reliance under long-tailed training, and clarifies the underlying limitations.

**Why a balanced model is a reasonable semantic reference.** Our diagnosis is based on the hypothesis that balanced training tends to produce explanations that are closer to class-relevant semantics. The key reason is supervision diversity: when each class has abundant and varied samples, spurious correlations (e.g., recurring contexts or textures) are less consistent across examples, and thus less reliable for minimizing the empirical risk. As a result, the model is pressured to rely on evidence that persists across diverse conditions, which is more likely to correspond to stable class-related cues. In contrast, long-tailed supervision reduces diversity for tail classes: a small number of examples often share similar acquisition conditions or contexts, making spurious cues highly predictive and therefore attractive shortcuts. Under this viewpoint, the balanced model provides a practical reference for "what evidence is likely to be used when shortcuts are less rewarded".

**Why deviations in CAMs can indicate shortcut reliance.** CAMs localize spatial evidence that contributes to a class score, providing an interpretable proxy for the model's reliance on different regions. For the same input $(x, y)$, if the long-tailed model learns a shortcut for class $y$, its evidence distribution may shift toward shortcut-correlated regions that are less stable or less semantically tied to the object. This shift results in a measurable discrepancy between the long-tailed CAM and the balanced reference CAM. Therefore, low cosine similarity between $\bar{M}_y^{lt}(x)$ and $\bar{M}_y^{bal}(x)$ is consistent with shortcut-dominated attention, whereas high similarity suggests the long-tailed model relies on evidence closer to the balanced reference.

**Role of normalization and cosine similarity.** We normalize CAMs to reduce scale effects (e.g., different magnitudes induced by logit scale or activation strength), and treat them as spatial evidence distributions. Cosine similarity then measures alignment in evidence *patterns* rather than absolute magnitude, making the metric more robust to global rescaling. Moreover, cosine similarity is sensitive to systematic spatial shifts: when evidence relocates from one region type to another (e.g., from object-supporting regions to shortcut-correlated regions), the vector directions diverge and similarity decreases.

**What this metric captures (and what it does not).** Our metric is designed to capture *relative changes* in spatial evidence allocation induced by long-tailed training. It is not intended to claim that the balanced CAM is a perfect "ground-truth" explanation, nor that every deviation is necessarily a shortcut. For example, architecture capacity, optimization noise, or class confusion can also affect CAM patterns. Nevertheless, in aggregate across samples and frequency groups, we expect long-tailed training to induce systematic deviations concentrated on tail classes if shortcut reliance is the dominant mechanism, which is precisely what we observe.

**Limitations.** Our CAM-similarity diagnosis is a *proxy* for shortcut reliance and has several limitations. First, the balanced reference CAM is not a ground-truth explanation: even under balanced training, CAMs may still be imperfect and can highlight discriminative parts rather than the full object, so deviations should be interpreted as *relative shifts* in evidence allocation rather than definitive proof of shortcuts. Second, low similarity does not uniquely imply shortcut learning—it may also arise from class confusion, optimization noise, or different local minima, especially for visually similar classes where the same image can admit multiple plausible evidence patterns. Third, CAM quality depends on the chosen architecture and CAM variant; for some backbones, CAMs can be spatially coarse or saturate due to feature-map resolution, which may blur fine-grained differences and weaken the interpretability of similarity. Finally, the metric assumes comparable training recipes and model capacity across the balanced and long-tailed runs; large differences in augmentation, regularization, or convergence can introduce confounding factors, making CAM discrepancies harder to attribute solely to long-tailed supervision. Despite these caveats, we find the trend is highly consistent across frequency groups, supporting its usefulness as an empirical indicator of shortcut-dominated attention in the long-tailed regime.

**Why this proxy is still useful for SRCD.** Importantly, our method does not require access to balanced training at deployment; the balanced reference is used only to diagnose the shortcut phenomenon. SRCD itself operates without any balanced reference model. The diagnostic result mainly motivates the use of head-class proxies: since head classes in long-tailed training enjoy richer supervision (closer to the balanced regime), their CAMs can serve as a practical source of shortcut-resistant evidence for tail classes.

## B.2. Additional Discussion on the Rationale of $\mathcal{E}_h$ and $\mathcal{E}_n$

Here, we provide a detailed rationale for why the spatial coherence energy $\mathcal{E}_h$ and the evidence concentration score $\mathcal{E}_n$ are suitable for detecting and mitigating shortcut learning in long-tailed recognition tasks, and discuss the assumptions and limitations of these measures.

**Why spatial coherence is important.** In long-tailed datasets, tail-class examples often share a similar contextual background, such as the same background textures, environments, or co-occurring objects, which can lead to shortcut learning. In contrast, head-class examples tend to be more diverse, with different object views, lighting conditions, and backgrounds, forcing the model to focus on the object itself. The spatial coherence energy $\mathcal{E}_h$ penalizes fragmented and disconnected activations, which are commonly seen in shortcut-driven explanations. Such fragmented patterns are often a result of relying on incidental background cues rather than stable, object-centric features.

By using Laplacian-based energy, $\mathcal{E}_h$ effectively captures the smoothness of the CAM, enforcing that evidence should be spatially coherent. This is particularly critical in long-tailed scenarios, where the risk of shortcut learning is higher, and background regions may dominate the tail-class CAMs. As a result, $\mathcal{E}_h$ encourages the model to focus on contiguous, stable regions that are more likely to correspond to meaningful object representations.

**The role of evidence concentration.** Tail classes in long-tailed datasets often have fewer examples, and these examples may be dominated by background context rather than the object itself. In these cases, the model may learn to focus on background cues (e.g., ocean or sky) instead of the actual object. The evidence concentration score, $\mathcal{E}_n$, addresses this issue by quantifying how concentrated the evidence is in a small region. A higher concentration score means the model is focusing its attention on a smaller, more discriminative region, which is typically associated with better object-centric explanations.

For example, in datasets like ImageNet, most objects are typically centered and well-defined, and background cues are less important. In such cases, the concentration score helps the model focus on the object itself, thus avoiding distractions from the background. By contrast, in long-tailed datasets where tail-class examples often share similar background features, the concentration score helps to suppress the model's reliance on large, diffuse background regions.

**Why these metrics work together.** While $\mathcal{E}_h$ and $\mathcal{E}_n$ capture different failure modes of shortcut learning, they complement each other in controlling both fragmentation and diffuse attention. A CAM may be compact but fragmented (e.g., a few isolated peaks in different locations), which is penalized by $\mathcal{E}_h$ but not necessarily by $\mathcal{E}_n$. On the other hand, a CAM may be smooth but overly spread across a large area (e.g., background activation covering a large portion of the image), which is penalized by $\mathcal{E}_n$ but may not yield a large coherence energy.

By combining both metrics, SRCD ensures that the final teacher CAM is both coherent and concentrated, making it more likely to reflect object-centric features rather than background or texture cues that may be prevalent in tail classes. These complementary aspects make SRCD a more reliable method for regularizing tail-class predictions.

**Dataset characteristics and generalization.** The effectiveness of $\mathcal{E}_h$ and $\mathcal{E}_n$ is particularly evident in datasets where background cues are prevalent, especially in long-tailed distributions. In datasets like ImageNet (Liu et al., 2019), where most classes are centered on an object and background context is less important, the combination of these metrics guides the model to focus on the object itself rather than on irrelevant background areas. In long-tailed data, however, the object-centric representations in tail classes are weaker, and background cues dominate. By using these metrics to prioritize head-class-like behavior, SRCD helps suppress shortcut reliance in tail classes.

Furthermore, we observe that these metrics generalize well to various long-tailed benchmarks. The assumption is that head classes, with more diverse supervision, have learned object-focused features. Therefore, SRCD leverages the better generalization of head-class explanations and transfers them to the tail classes, where there is a higher risk of shortcut learning.

**Limitations of the approach.** Despite their usefulness, $\mathcal{E}_h$ and $\mathcal{E}_n$ are not perfect proxies for object-centric semantics. There are a few cases where the metrics may penalize valid evidence:

- **Large objects or irregular shapes:** For some objects, the relevant evidence may naturally be spread over large or irregular regions (e.g., a large animal or vehicle), which might be penalized by $\mathcal{E}_n$ for being diffuse, even though it is semantically important.

- **Disconnected object parts:** Some objects may have parts that are naturally disconnected (e.g., thin structures), which could result in a fragmented CAM, and thus a higher $\mathcal{E}_h$. In such cases, the metric may incorrectly penalize valid object evidence.

- **Fine-grained categories:** In fine-grained classification tasks, where classes are very similar (e.g., different species of birds), $\mathcal{E}_h$ and $\mathcal{E}_n$ may not always perfectly distinguish between legitimate object features and spurious correlations, requiring further fine-tuning of the criteria.

**Practical considerations.** We compute both $\mathcal{E}_h$ and $\mathcal{E}_n$ on normalized CAMs $\bar{M}$ to avoid scale effects that could be introduced by different activation magnitudes across classes. This normalization ensures that the metrics focus on the spatial distribution of attention rather than absolute scale. The Laplacian matrix $L$ is computed once per input resolution and reused throughout training, which makes the computation efficient. Additionally, the 4-neighbor grid used in the Laplacian has been found to work well in most datasets, but alternative choices, such as 8-neighbor grids, could be explored for more complex tasks.

**Conclusion.** In summary, $\mathcal{E}_h$ and $\mathcal{E}_n$ are designed to capture different aspects of shortcut reliance in long-tailed learning. By enforcing spatial coherence and evidence concentration, SRCD effectively suppresses shortcut-driven behavior in tail classes. These metrics are not perfect but provide a reliable mechanism to regularize tail-class predictions, especially in datasets where background cues dominate. Despite certain limitations, SRCD remains a robust tool for improving the generalization of models trained on long-tailed datasets.

### B.3. Discussion on Head-Proxy CAM Supervision

SRCD uses head-proxy CAMs as a shortcut-resistant supervision signal for tail classes. We emphasize that these proxies are not intended to provide perfect semantic matching or complete part-level supervision for the target tail object. In particular, we do not assume that different head proxies must cover mutually complementary object parts, nor do we require each proxy CAM to strictly align with the full target object. Even for head classes with abundant training samples, obtaining complete and semantically correct part-level CAMs is difficult. Therefore, enforcing such a requirement on tail examples, where supervision is scarce and often biased, would be overly strong.

The goal of SRCD is different from explicit part decomposition. Rather than constructing a pixel-level or part-level ground-truth explanation, SRCD aims to build a teacher CAM that is more stable and less affected by shortcut cues than the original tail-class CAM. This distinction is important: a useful teacher does not need to perfectly cover every object part. It can still provide effective supervision as long as it is less shortcut-prone than the tail CAM learned from scarce and biased data. In this sense, SRCD uses head-proxy CAMs as a regularization signal to encourage tail explanations to move away from background or context shortcuts.

Semantic mismatch between head proxies and tail classes may also occur. SRCD does not assume that every selected proxy is reliable. Instead, it aggregates multiple candidate CAMs using energy-based weights based on spatial coherence and evidence concentration. This design suppresses fragmented, noisy, or overly diffuse proxy CAMs and favors candidates that provide more stable spatial evidence. As a result, the teacher is not determined by a single potentially mismatched proxy, but by a weighted combination of candidates that are structurally more consistent with shortcut-resistant attention.

To further examine the effect of semantic mismatch, we conduct an additional analysis on iNaturalist using its taxonomy labels, such as Actinopterygii, Amphibia, Animalia, Arachnida, Aves, Insecta, Mammalia, and Plantae. Specifically, we identify a mismatched subset of tail samples whose selected head proxies are taxonomically unrelated to the target tail class. With BCL as the baseline, 32.13% of tail samples fall into this mismatched subset. As shown in Table 4, SRCD still improves the accuracy on these samples from 67.25 to 69.88. This result suggests that SRCD does not strictly rely on exact semantic matching between the head proxy and the tail target. Even when the selected proxy is taxonomically unrelated, its CAM can still provide useful shortcut-resistant spatial evidence, as long as it is less shortcut-prone than the original tail-class CAM.

*Table 4.* Accuracy (%) on the semantically mismatched subset of iNaturalist tail samples, where the selected head proxies are taxonomically unrelated to the target tail classes.

| Method | Acc. |
|---|---|
| BCL | 67.25 |
| BCL + SRCD | 69.88 |

One possible alternative is to explicitly enforce diversity across proxies so that different candidates attend to different object parts. However, this would require additional assumptions and computational costs, such as a larger top-$K$ candidate set, fine-grained pixel-level comparisons among CAMs, and extra diversity or structural constraints. More importantly, without dense annotations, it is difficult to verify whether the resulting regions are truly correct, complete, and complementary. This difficulty is even more pronounced for irregular objects, such as insects, where object parts can be small, deformable, or visually ambiguous.

Therefore, SRCD should be understood as an evidence-level shortcut suppression method rather than a part-level decomposition method. Its objective is to provide tail classes with a less shortcut-prone CAM teacher, not a perfect semantic or part-level explanation. The empirical improvements across datasets and baselines suggest that this form of shortcut-resistant CAM supervision is sufficient to improve tail-class generalization.

This design is also related to recent observations in open-set noisy-label learning that out-of-set or non-target samples are not necessarily purely harmful; when properly exploited, they can provide useful auxiliary information for improving class separation (Wan et al., 2024). Similarly, SRCD does not require head proxies to be semantically identical to the tail label, but uses them as candidate spatial explanations whose reliability is further filtered by energy-based weighting.

### B.4. Long-Tailed-Specific Design of SRCD

SRCD is designed specifically for the long-tailed setting. The key challenge is not only class imbalance itself, but also the asymmetric shortcut vulnerability induced by such imbalance. Tail classes, under scarce and biased supervision, are more likely to develop shortcut-dominated attention, whereas head classes tend to learn more stable object-centric evidence due to richer and more diverse training samples.

Motivated by this long-tailed-specific observation, SRCD uses head-side explanations to regularize tail-class attention during training. Therefore, the role of CAM distillation in SRCD is not generic. Instead, the distillation is constructed around the head-tail structure: the teacher is built from head-class proxies, the correction is applied specifically to tail classes, and the regularization strength is scaled by class rarity. In this sense, SRCD is a long-tailed-motivated mechanism for correcting shortcut-dominated tail attention, rather than a generic CAM distillation strategy directly applied to long-tailed data.

## C. Proofs and Extended Discussion

### C.1. Notation and Preliminaries

**Normalized CAM.** For an input image $x$ and class $y$, let $\bar{M}_y(x) \in \mathbb{R}^{H \times W}$ denote the *normalized* CAM:

$$\bar{M}_y(x)_u \geq 0, \qquad \sum_{u \in \Omega} \bar{M}_y(x)_u = 1, \tag{9}$$

where $\Omega$ indexes the spatial grid (with $|\Omega| = N = HW$). Hence, $\bar{M}_y(x)$ can be viewed as a discrete probability distribution over spatial locations.

**Shortcut/background region.** We assume the spatial domain $\Omega$ can be partitioned into an object-related region $\Omega_{\text{obj}}$ and a shortcut/background region $\Omega_{\text{sc}}$:

$$\Omega = \Omega_{\text{obj}} \cup \Omega_{\text{sc}}, \qquad \Omega_{\text{obj}} \cap \Omega_{\text{sc}} = \varnothing. \tag{10}$$

This partition is *conceptual*: the analysis does not require explicit masks during training; it only serves to formalize the shortcut mechanism.

**Shortcut-region attention mass.** We quantify shortcut reliance of any normalized CAM $\bar{M}$ as:

$$\mathcal{A}_{\text{sc}}(\bar{M}) = \sum_{u \in \Omega_{\text{sc}}} \bar{M}_u \in [0, 1]. \tag{11}$$

Smaller $\mathcal{A}_{\text{sc}}$ indicates more object-centric explanations; larger $\mathcal{A}_{\text{sc}}$ indicates stronger shortcut/background attention.

**Teacher construction.** Let $\mathcal{H}(x)$ denote the set of head-proxy candidates used to form the teacher CAM. SRCD defines:

$$\bar{M}_T(x) = \sum_{c \in \mathcal{H}(x)} w_c(x)\,\bar{M}_c(x), \qquad w_c(x) = \frac{\exp\big(\mathcal{U}(\bar{M}_c(x))/\tau\big)}{\sum_{c' \in \mathcal{H}(x)} \exp\big(\mathcal{U}(\bar{M}_{c'}(x))/\tau\big)}, \tag{12}$$

where $\tau > 0$ is a temperature and the utility is

$$\mathcal{U}(\bar{M}) = -\mathcal{E}_h(\bar{M}) + \mathcal{E}_n(\bar{M}). \tag{13}$$

Intuitively, $\mathcal{E}_h$ penalizes spatial fragmentation (favoring coherent evidence), while $\mathcal{E}_n$ rewards evidence concentration (favoring object-centric evidence).

**CAM distillation loss.** Given a tail sample $(x, y)$, SRCD distills the teacher explanation into the tail-class CAM:

$$\mathcal{L}_{\text{SRCD}}(x, y) = \gamma_y \big\| \bar{M}_y(x) - \bar{M}_T(x) \big\|_2^2, \tag{14}$$

where $\gamma_y$ increases with tail scarcity to emphasize shortcut mitigation for low-data classes.

## C.2. Proof of Theorem 4.1 (Entropy-Regularized Utility Maximization)

We restate the theorem in a self-contained form.

**Theorem C.1** (Entropy-regularized utility maximization). *Let $\{\mathcal{U}_c\}_{c=1}^K$ be utilities and $\Delta = \{w \in \mathbb{R}^K : w_c \geq 0, \sum_{c=1}^K w_c = 1\}$. For $\tau > 0$, the optimization*

$$\max_{w \in \Delta} \quad J(w) = \sum_{c=1}^K w_c \mathcal{U}_c - \tau \sum_{c=1}^K w_c \log w_c \tag{15}$$

*has a unique maximizer*

$$w_c^\star = \frac{\exp(\mathcal{U}_c/\tau)}{\sum_{j=1}^K \exp(\mathcal{U}_j/\tau)}. \tag{16}$$

**Step 1: Strict concavity and uniqueness.** The first term $\sum_c w_c \mathcal{U}_c$ is linear in $w$. The entropy term $-\sum_c w_c \log w_c$ is strictly concave over the interior of $\Delta$ (and concave over the whole simplex). Thus, $J(w)$ is strictly concave on $\Delta$, implying a unique global maximizer.

**Step 2: Lagrangian and KKT conditions.** Introduce a Lagrange multiplier $\lambda$ for the simplex constraint:

$$\mathcal{L}(w, \lambda) = \sum_{c=1}^K w_c \mathcal{U}_c - \tau \sum_{c=1}^K w_c \log w_c + \lambda \Big( \sum_{c=1}^K w_c - 1 \Big). \tag{17}$$

For an interior optimum ($w_c > 0$), the stationarity condition is:

$$\frac{\partial \mathcal{L}}{\partial w_c} = \mathcal{U}_c - \tau(\log w_c + 1) + \lambda = 0. \tag{18}$$

Rearranging yields:

$$\log w_c = \frac{\mathcal{U}_c + \lambda - \tau}{\tau} \quad \Longrightarrow \quad w_c = \exp\Big(\frac{\mathcal{U}_c}{\tau}\Big) \cdot \exp\Big(\frac{\lambda - \tau}{\tau}\Big). \tag{19}$$

Let $Z = \sum_{j=1}^K \exp(\mathcal{U}_j/\tau)$. Enforcing $\sum_c w_c = 1$ gives:

$$\exp\Big(\frac{\lambda - \tau}{\tau}\Big) = \frac{1}{Z} \quad \Longrightarrow \quad w_c^\star = \frac{\exp(\mathcal{U}_c/\tau)}{Z}. \tag{20}$$

This proves (16).

**Interpretation (why this is principled).** Eq. (15) formalizes a standard "exploitation–exploration" trade-off:

- $\sum_c w_c \mathcal{U}_c$ pushes the teacher to select candidates with *high utility* (low fragmentation $\mathcal{E}_h$ and high concentration $\mathcal{E}_n$).

- The entropy penalty $-\sum_c w_c \log w_c$ prevents brittle hard selection, improving robustness when multiple candidates have similar quality.

- The temperature $\tau$ controls smoothness: $\tau \to 0$ approaches hard max selection; larger $\tau$ approaches uniform averaging.

**Useful equivalent form (free-energy).** Define the log-sum-exp value:

$$\Phi(\mathcal{U}) = \tau \log \sum_{c=1}^K \exp(\mathcal{U}_c/\tau). \tag{21}$$

One can show:

$$\max_{w \in \Delta} \Big( \sum_c w_c \mathcal{U}_c - \tau \sum_c w_c \log w_c \Big) = \Phi(\mathcal{U}), \tag{22}$$

and $w^\star = \nabla_{\mathcal{U}} \Phi(\mathcal{U})$ equals the softmax weights. This view is useful when discussing stability and smoothness properties of the teacher.

### C.3. Proof of Theorem 4.2 (Distillation Dynamics)

We restate Theorem 4.2 and provide a detailed derivation.

**Theorem C.2** (Monotone shortcut-mass contraction). *Consider the gradient flow minimizing $F(\bar{M}) = \|\bar{M} - \bar{M}_T\|_2^2$:*

$$\frac{d}{dt}\bar{M}(t) = -\nabla_{\bar{M}} F(\bar{M}(t)). \tag{23}$$

*Then the solution is $\bar{M}(t) = \bar{M}_T + e^{-2t}(\bar{M}(0) - \bar{M}_T)$, and*

$$\mathcal{A}_{\mathrm{sc}}(\bar{M}(t)) = \mathcal{A}_{\mathrm{sc}}(\bar{M}_T) + e^{-2t}\big(\mathcal{A}_{\mathrm{sc}}(\bar{M}(0)) - \mathcal{A}_{\mathrm{sc}}(\bar{M}_T)\big). \tag{24}$$

*If $\mathcal{A}_{\mathrm{sc}}(\bar{M}(0)) > \mathcal{A}_{\mathrm{sc}}(\bar{M}_T)$ then $\mathcal{A}_{\mathrm{sc}}(\bar{M}(t))$ decreases monotonically to $\mathcal{A}_{\mathrm{sc}}(\bar{M}_T)$.*

**Step 1: Compute the gradient.** Let $F(\bar{M}) = \|\bar{M} - \bar{M}_T\|_2^2 = (\bar{M} - \bar{M}_T)^\top (\bar{M} - \bar{M}_T)$. Then:

$$\nabla_{\bar{M}} F(\bar{M}) = 2(\bar{M} - \bar{M}_T). \tag{25}$$

**Step 2: Solve the ODE.** Plug into (23):

$$\frac{d}{dt}\bar{M}(t) = -2(\bar{M}(t) - \bar{M}_T). \tag{26}$$

This is a linear ODE. Let $\Delta(t) = \bar{M}(t) - \bar{M}_T$. Then:

$$\frac{d}{dt}\Delta(t) = -2\Delta(t) \quad\Longrightarrow\quad \Delta(t) = e^{-2t}\Delta(0) = e^{-2t}(\bar{M}(0) - \bar{M}_T), \tag{27}$$

hence

$$\bar{M}(t) = \bar{M}_T + e^{-2t}(\bar{M}(0) - \bar{M}_T). \tag{28}$$

**Step 3: Shortcut-mass transport follows from linearity.** Define the indicator vector $\mathbf{1}_{\Omega_{\mathrm{sc}}}$ (with 1 on $\Omega_{\mathrm{sc}}$ and 0 elsewhere). Then:

$$\mathcal{A}_{\mathrm{sc}}(\bar{M}) = \langle \mathbf{1}_{\Omega_{\mathrm{sc}}}, \bar{M}\rangle. \tag{29}$$

Taking inner products of (28) with $\mathbf{1}_{\Omega_{\mathrm{sc}}}$ yields:

$$\mathcal{A}_{\mathrm{sc}}(\bar{M}(t)) = \langle \mathbf{1}_{\Omega_{\mathrm{sc}}}, \bar{M}_T\rangle + e^{-2t}\langle \mathbf{1}_{\Omega_{\mathrm{sc}}}, \bar{M}(0) - \bar{M}_T\rangle \tag{30}$$

$$= \mathcal{A}_{\mathrm{sc}}(\bar{M}_T) + e^{-2t}\big(\mathcal{A}_{\mathrm{sc}}(\bar{M}(0)) - \mathcal{A}_{\mathrm{sc}}(\bar{M}_T)\big), \tag{31}$$

which proves (24). Monotonicity follows immediately if $\mathcal{A}_{\mathrm{sc}}(\bar{M}(0)) > \mathcal{A}_{\mathrm{sc}}(\bar{M}_T)$.

**Discrete-time analogue (what SGD is doing).** If we take a simple gradient step with step size $\eta \in (0,1)$:

$$\bar{M}^{(t+1)} = \bar{M}^{(t)} - \eta\,\nabla_{\bar{M}}\|\bar{M}^{(t)} - \bar{M}_T\|_2^2 = (1 - 2\eta)\bar{M}^{(t)} + 2\eta\,\bar{M}_T. \tag{32}$$

Thus, each step is an explicit interpolation toward $\bar{M}_T$. Applying $\mathcal{A}_{\mathrm{sc}}(\cdot)$ gives:

$$\mathcal{A}_{\mathrm{sc}}(\bar{M}^{(t+1)}) = (1 - 2\eta)\mathcal{A}_{\mathrm{sc}}(\bar{M}^{(t)}) + 2\eta\,\mathcal{A}_{\mathrm{sc}}(\bar{M}_T), \tag{33}$$

so shortcut mass decreases whenever the teacher has smaller shortcut mass. This provides an optimization-level explanation beyond a static inequality bound.

### C.4. Why the Teacher is Shortcut-Resistant: Utility Alignment and Consequences

Theorems above show that distillation *will* transport shortcut mass toward the teacher. We now justify why the proposed teacher is expected to be shortcut-resistant.

### C.4.1. A WEAK ALIGNMENT ASSUMPTION

We adopt a mild and interpretable condition: the shortcut mass is negatively aligned with utility.

**Assumption C.3** (Utility–shortcut alignment). There exist constants $\alpha \in \mathbb{R}$ and $\beta > 0$ such that for all candidates $c \in \mathcal{H}(x)$,

$$\mathcal{A}_{\mathrm{sc}}(\bar{M}_c(x)) \leq \alpha - \beta \mathcal{U}(\bar{M}_c(x)). \tag{34}$$

**Why this assumption is reasonable.** The utility $\mathcal{U} = -\mathcal{E}_h + \mathcal{E}_n$ favors *coherent* and *concentrated* evidence:

- Background shortcuts typically appear as scattered activations across many regions (high fragmentation), yielding larger $\mathcal{E}_h$ and lower utility.

- Shortcut-driven CAMs are often diffuse (low concentration), yielding smaller $\mathcal{E}_n$ and lower utility.

Thus, higher $\mathcal{U}$ tends to correlate with lower background mass $\mathcal{A}_{\mathrm{sc}}$.

### C.4.2. TEACHER SHORTCUT MASS BOUND UNDER ALIGNMENT

Since $\bar{M}_T = \sum_c w_c \bar{M}_c$ and $\mathcal{A}_{\mathrm{sc}}$ is linear:

$$\mathcal{A}_{\mathrm{sc}}(\bar{M}_T) = \sum_c w_c \, \mathcal{A}_{\mathrm{sc}}(\bar{M}_c). \tag{35}$$

Applying Assumption C.3:

$$\mathcal{A}_{\mathrm{sc}}(\bar{M}_T) \leq \sum_c w_c(\alpha - \beta \mathcal{U}_c) = \alpha - \beta \sum_c w_c \mathcal{U}_c, \tag{36}$$

where $\mathcal{U}_c = \mathcal{U}(\bar{M}_c)$.

**Key implication.** Eq. (36) shows the teacher shortcut mass is upper-bounded by *negative expected utility*. Therefore, any mechanism that increases $\sum_c w_c \mathcal{U}_c$ will reduce (or tightly control) the teacher shortcut mass.

### C.4.3. SOFTMAX WEIGHTS MAXIMIZE EXPECTED UTILITY (WITH ROBUSTNESS)

By Theorem 4.1, the softmax weights are the unique maximizer of

$$\sum_c w_c \mathcal{U}_c - \tau \sum_c w_c \log w_c.$$

This yields two desirable properties:

- **Low-shortcut preference:** candidates with larger $\mathcal{U}_c$ receive exponentially larger weights $w_c$.

- **Robustness:** the entropy term avoids collapsing onto a single candidate when utilities are close or noisy.

Combining with (36), the proposed teacher aggregation is systematically biased toward lower shortcut mass.

### C.4.4. FROM TEACHER QUALITY TO TAIL SHORTCUT SUPPRESSION

If the aggregated teacher satisfies

$$\mathcal{A}_{\mathrm{sc}}(\bar{M}_T) < \mathcal{A}_{\mathrm{sc}}(\bar{M}_y), \tag{37}$$

then Theorem 4.2 implies that distillation decreases the tail shortcut mass monotonically (and exponentially under the gradient-flow view). This establishes the end-to-end shortcut mitigation mechanism:

utility-biased teacher $\Rightarrow$ teacher is more object-centric $\Rightarrow$ $L_2$ distillation transports mass away from shortcuts.

### C.5. Technical Details of $\mathcal{E}_h$: Graph Laplacian Coherence Energy

We provide a detailed explanation of $\mathcal{E}_h(\bar{M}) = \bar{M}^\top L \bar{M}$ and why it captures spatial coherence.

### C.5.1. WHAT IS $L$ AND WHY IT IS USED

Let $G = (V, \mathcal{E})$ be the 4-neighbor grid graph over pixels, where $V = \Omega$ and $(i, j) \in \mathcal{E}$ if pixels $i$ and $j$ are adjacent. Let $G \in \mathbb{R}^{N \times N}$ be the adjacency matrix ($G_{ij} = 1$ if $(i, j) \in \mathcal{E}$, else 0), and $D$ be the degree matrix ($D_{ii} = \sum_j G_{ij}$). The (combinatorial) graph Laplacian is

$$L = D - G. \tag{38}$$

**Dirichlet energy interpretation.**   For any vector $z \in \mathbb{R}^N$:

$$z^\top L z = \frac{1}{2} \sum_{i,j} G_{ij}(z_i - z_j)^2 = \sum_{(i,j) \in \mathcal{E}} (z_i - z_j)^2. \tag{39}$$

Thus, $z^\top L z$ penalizes local discontinuities across neighboring pixels.

**Why it captures "shortcut fragmentation".**   CAMs driven by shortcut/background cues often show scattered, high-frequency activations across diverse regions. Such patterns have large neighbor differences $(\bar{M}_i - \bar{M}_j)^2$, increasing $\bar{M}^\top L \bar{M}$. In contrast, object-centric evidence tends to form connected blobs with smooth interiors and fewer abrupt changes, yielding smaller coherence energy.

### C.5.2. SPECTRAL VIEW: HIGH-FREQUENCY SUPPRESSION

Since $L$ is symmetric positive semidefinite, it admits $L = U \Lambda U^\top$, where $\Lambda = \text{diag}(\lambda_1, \dots, \lambda_N)$ with $0 = \lambda_1 \leq \cdots \leq \lambda_N$. Then:

$$\bar{M}^\top L \bar{M} = \sum_{k=1}^{N} \lambda_k \langle u_k, \bar{M} \rangle^2. \tag{40}$$

Large eigenvalues correspond to high-frequency graph components. Therefore, minimizing $\mathcal{E}_h$ suppresses high-frequency oscillations in CAMs—precisely the type of spatial noise often induced by shortcuts.

### C.5.3. DISCUSSION: COHERENCE ENERGY VS. "PURE CONNECTIVITY"

A purely connectivity-based metric (e.g., number of connected components after thresholding) is non-differentiable and sensitive to thresholds. In contrast, $\bar{M}^\top L \bar{M}$ is:

- **Differentiable and stable:** smoothly changes under small perturbations.

- **Scale-aware:** penalizes fine-grained fragmentation without requiring binarization.

- **Computationally cheap:** computable via local neighbor differences on the grid.

These properties make $\mathcal{E}_h$ a practical surrogate for shortcut-related fragmentation in CAM space.

### C.6. Technical Details of $\mathcal{E}_n$: Evidence Concentration Score

We discuss why $\mathcal{E}_n$ is a suitable measure for discouraging diffuse, background-spread evidence.

### C.6.1. A NORMALIZED CONCENTRATION SCORE

Let $N = |\Omega|$ and $\bar{M} \geq 0$ with $\|\bar{M}\|_1 = 1$ (by normalization). Consider the Hoyer-style normalized score:

$$\mathcal{E}_n(\bar{M}) = \frac{\sqrt{N} - \|\bar{M}\|_1 / \|\bar{M}\|_2}{\sqrt{N} - 1} = \frac{\sqrt{N} - 1/\|\bar{M}\|_2}{\sqrt{N} - 1}, \tag{41}$$

which lies in $[0, 1]$.

**Extreme cases (sanity check).**

- **Uniform/diffuse CAM:** $\bar{M}_u = \frac{1}{N}$ for all $u$. Then $\|\bar{M}\|_2 = 1/\sqrt{N}$ and $\mathcal{E}_n(\bar{M}) = 0$.

- **One-hot CAM:** all mass on one pixel. Then $\|\bar{M}\|_2 = 1$ and $\mathcal{E}_n(\bar{M}) = 1$.

Thus, larger $\mathcal{E}_n$ corresponds to more concentrated evidence.

**Why concentration discourages shortcuts.** Shortcut reliance often manifests as evidence spread across large background regions (e.g., texture, context), increasing diffusion and reducing peakiness. Rewarding concentration therefore reduces the probability mass assigned to broad, non-object background areas.

### C.6.2. SCALE MATCHING WITH $\mathcal{E}_h$

In practice, $\mathcal{E}_h$ can have a larger numerical range than $\mathcal{E}_n \in [0, 1]$. Using $\mathcal{U} = -\mathcal{E}_h + \mathcal{E}_n$ implicitly assumes the two terms are roughly comparable. A common and robust alternative is:

$$\mathcal{U}(\bar{M}) = -\lambda_h \mathcal{E}_h(\bar{M}) + \lambda_n \mathcal{E}_n(\bar{M}), \tag{42}$$

with $\lambda_h, \lambda_n > 0$. This preserves the analysis and typically improves stability of teacher selection.

### C.7. Additional Bounds: Shortcut Mass Gap Controlled by Distillation Distance

Although our main text focuses on the transport dynamics, it is also useful to provide a static inequality bound.

**Lemma C.4** (Shortcut-mass gap bound). *For any normalized CAMs $\bar{M}, \bar{M}_T$,*

$$\left| \mathcal{A}_{\mathrm{sc}}(\bar{M}) - \mathcal{A}_{\mathrm{sc}}(\bar{M}_T) \right| \le \sqrt{|\Omega_{\mathrm{sc}}|} \, \|\bar{M} - \bar{M}_T\|_2. \tag{43}$$

**Proof.** Let $\Delta = \bar{M} - \bar{M}_T$. Then

$$\mathcal{A}_{\mathrm{sc}}(\bar{M}) - \mathcal{A}_{\mathrm{sc}}(\bar{M}_T) = \sum_{u \in \Omega_{\mathrm{sc}}} \Delta_u.$$

By Cauchy–Schwarz:

$$\left| \sum_{u \in \Omega_{\mathrm{sc}}} \Delta_u \right| \le \sqrt{|\Omega_{\mathrm{sc}}|} \cdot \sqrt{\sum_{u \in \Omega_{\mathrm{sc}}} \Delta_u^2} \le \sqrt{|\Omega_{\mathrm{sc}}|} \, \|\Delta\|_2.$$

$\square$

**Discussion.** Lemma C.4 directly connects the optimization target $\|\bar{M} - \bar{M}_T\|_2^2$ to shortcut mass discrepancy. It complements the transport view by providing a simple worst-case guarantee even without analyzing learning dynamics.

### C.8. Why SRCD Should be Enabled After Head Classes Become Reliable

SRCD distillation is only beneficial if the teacher is indeed shortcut-resistant. This requirement motivates a late-start (warm-up) schedule.

**Early-stage risk.** In early epochs, head-class CAMs may not be fully semantic and can still attend to spurious background cues. In this regime, utilities $\mathcal{U}(\bar{M}_c)$ are noisy, and teacher aggregation may fail to reduce shortcut mass, potentially amplifying shortcuts through distillation.

**Late-stage advantage.** After the model learns strong head semantics, head-proxy CAMs become more coherent and concentrated: $\mathcal{E}_h$ decreases and $\mathcal{E}_n$ increases, hence $\mathcal{U}$ becomes reliably higher for object-centric evidence. Teacher construction then becomes shortcut-resistant, and Theorem 4.2 guarantees monotone shortcut-mass reduction.

**Compatibility with fine-tuning.** This naturally aligns with long-tailed training protocols that apply re-balancing in a second stage. SRCD can be introduced during the fine-tuning phase as a plug-and-play explanation regularizer on top of any re-balancing objective, without modifying architectures.

### C.9. Extended Discussion and Practical Takeaways

**(1) What exactly is "shortcut" in our theory?**   Shortcut learning is modeled as excessive CAM mass on $\Omega_{\text{sc}}$. This focuses the theory on *where* the model draws evidence, rather than only how confident it is. It matches the empirical failure mode where tail classes rely on correlated backgrounds.

**(2) Why CAM distillation (not feature distillation) is the right lever?**   Feature alignment may still permit shortcut reliance if background cues dominate internal representations. By contrast, CAM alignment targets the *spatial evidence* directly, enforcing object-centric explanations. The transport identity (24) makes this effect explicit.

**(3) Why combine coherence and concentration?**   Either term alone is insufficient:

- Low $\mathcal{E}_h$ alone can be achieved by smooth but *diffuse* CAMs covering background.

- High $\mathcal{E}_n$ alone can be achieved by *peaky* but fragmented evidence.

Using $\mathcal{U} = -\mathcal{E}_h + \mathcal{E}_n$ favors CAMs that are simultaneously blob-like (coherent) and object-localized (concentrated), which better matches semantic evidence and discourages shortcut backgrounds.

**(4) Temperature $\tau$ and stability.**   Smaller $\tau$ performs near-hard selection of the top-utility candidate, which may be brittle when utilities are noisy. Larger $\tau$ averages more candidates, improving stability but weakening selectivity. Entropy-regularized utility maximization provides a clean interpretation for choosing $\tau$.

**(5) Why SRCD is especially effective for tail classes.**   Tail supervision is data-scarce, so learned explanations are more likely to drift toward shortcuts. SRCD introduces an additional signal—a shortcut-resistant teacher—and Theorem 4.2 shows this signal induces monotone shortcut-mass reduction. The tail-weight $\gamma_y$ further amplifies this correction.

**(6) What does the theory *not* assume?**   We do not assume availability of segmentation masks or explicit knowledge of $\Omega_{\text{sc}}$ during training. The region split is only for analysis, while the learning signal arises from coherence and concentration energies, which are computed directly from CAMs.

**(7) Summary.**   The appendix provides a complete shortcut-centric explanation of SRCD: teacher construction is a principled entropy-regularized utility maximizer (Section C.2), and $L_2$ CAM distillation induces an explicit contraction/transport dynamics that monotonically reduces shortcut attention mass toward a more object-centric teacher (Section C.3–C.4).

# D. Experimental Details

## D.1. Detailed Dataset Description

We leverage five benchmark datasets — CIFAR100-LT (Cao et al., 2019), ImageNet-LT (Liu et al., 2019), iNaturalist 2018 (Van Horn et al., 2018), and Places-LT (Liu et al., 2019) — to mimic real - world long - tailed class distributions. These datasets are characterized by severe class imbalance, as noted in previous studies (Zhang et al., 2023). Table 5 summarizes the data statistics of these datasets. Notably, CIFAR100-LT features three variants, each with a distinct imbalance ratio. The imbalance ratio is defined as $\frac{\max n_j}{\min n_j}$, where $n_j$ represents the number of data points in class $j$.

*Table 5.* Statistics of datasets.

| Dataset | # classes | # training data | # test data | imbalance ratio |
|---|---|---|---|---|
| CIFAR10-LT (Cao et al., 2019) | 10 | 50,000 | 10,000 | $\{100, 50, 10\}$ |
| CIFAR100-LT (Cao et al., 2019) | 100 | 50,000 | 10,000 | $\{100, 50, 10\}$ |
| ImageNet-LT (Liu et al., 2019) | 1,000 | 115,846 | 50,000 | 256 |
| iNaturalist 2018 (Van Horn et al., 2018) | 8,142 | 437,513 | 24,426 | 500 |
| Places-LT (Liu et al., 2019) | 365 | 62,500 | 36,500 | 996 |

**CIFAR10/100-LT.** CIFAR10/100-LT are long-tailed variants of CIFAR-10 and CIFAR-100 (Krizhevsky et al., 2009). CIFAR-10 contains 50,000 training images and 10,000 validation images of size ($32 \times 32$) from 10 classes, while CIFAR-100 contains 50,000 training images and 10,000 validation images of size ($32 \times 32$) from 100 classes. Following (Cao et al., 2019), we adopt the same long-tailed data construction for fair comparison. The imbalance ratio (IR) is defined as $\rho = N_{\max}/N_{\min}$, where $N_{\max}$ and $N_{\min}$ denote the numbers of samples in the most frequent and least frequent classes, respectively. Unless otherwise specified, we evaluate on $\rho \in \{100, 50, 10\}$ for both CIFAR10-LT and CIFAR100-LT.

**ImageNet-LT.** ImageNet-LT is a long-tailed version of ImageNet-2012 (Deng et al., 2009) introduced by (Liu et al., 2019). Following (Liu et al., 2019), we use the same construction by sampling training images according to a Pareto distribution with power $\gamma = 6$. ImageNet-LT contains $115.8K$ images from 1,000 categories, with an imbalance factor $\rho = N_{\max}/N_{\min} = 1280/5$.

**iNaturalist 2018.** iNaturalist 2018 (Van Horn et al., 2018) is a large-scale real-world long-tailed recognition benchmark with an extremely imbalanced class distribution. It contains $437.5K$ training images and $24.4K$ validation images from 8,142 categories. Its fine-grained categories further increase the recognition difficulty.

## D.2. Competitors

For a fair and controlled comparison, we apply our method on top of each baseline *without changing* its original architecture. All experiments follow the same training pipeline as the corresponding baseline, and we keep all optimization and training hyperparameters (e.g., learning rate schedule, optimizer, epochs, batch size) *identical* across methods. All experiments are conducted on four NVIDIA A100 GPUs. Moreover, SRCD is enabled only after a warm-up stage and is applied in the late training phase, when head-class representations become sufficiently reliable.

Unless otherwise specified, we fix the SRCD loss weight to $\beta = 0.6$ and set $K = 3$ in all experiments.

In our experiments, we benchmark our proposed method against a comprehensive suite of state-of-the-art algorithms. We categorize these baselines into two prevailing training paradigms: training from scratch on standard long-tailed benchmarks and fine-tuning foundation models.

### D.2.1. TRAINING FROM SCRATCH

For the standard training-from-scratch setting, we compare with the following representative methods, ordered by their release timeline and technical approach:

- BCL (Balanced Contrastive Learning) (Zhu et al., 2022): BCL identifies that standard contrastive learning is inherently biased towards head classes due to the dominance of majority samples in the mini-batch. To mitigate this, it introduces a class-averaging mechanism that equalizes the gradient contributions of different classes. Furthermore, it employs

a balanced temperature parameter to dynamically adjust the difficulty of negative samples, ensuring that the learned feature space remains discriminative for both head and tail categories.

- GCL (Gaussian Clouded Logit Adjustment) (Li et al., 2022b): GCL approaches the long-tailed problem from the perspective of uncertainty estimation. It observes that tail classes suffer from high epistemic uncertainty, which manifests as vague decision boundaries. By modeling the logits of each class as a Gaussian distribution (a "cloud") rather than a deterministic point, GCL enforces a calibration constraint that pushes the decision boundaries of tail classes outward, effectively compensating for the lack of samples with a probabilistic margin.

- FCC (Feature Clusters Compression) (Li et al., 2023): FCC focuses on optimizing the geometric structure of the feature space. It argues that head classes, with their abundant samples, tend to form expansive clusters with large intra-class variance, which "intrude" into the limited feature space available for tail classes. FCC proposes a cluster compression mechanism to explicitly constrain the variance of head classes, thereby reserving more embedding space for tail categories and reducing feature confusion.

- GLMC (Global and Local Mixture Consistency) (Du et al., 2023): GLMC combines the benefits of cumulative learning with consistency regularization. It utilizes a global branch to maintain a historical context of the class distribution (addressing the forgetting issue in streaming data) and a local branch to enforce consistency via Mixup augmentation. By aligning the predictions of these two branches, GLMC ensures that the model learns robust representations that are invariant to the sampling bias inherent in long-tailed data.

- H2T (Head-to-Tail Feature Fusion) (Li et al., 2024): H2T exploits the high-quality semantic representations learned from head classes to aid the recognition of tail classes. It designs a feature fusion module that transfers diverse semantic statistics (e.g., variance and prototypes) from head categories to augment the sparse representations of tail categories. This "Head-to-Tail" transfer effectively hallucinates diverse samples for the tail, enriching their feature distribution without requiring additional real data.

- BNS (Information-Preservable Two-Stage Learning) (Lin & Yuan, 2025): BNS revisits the classical two-stage decoupling paradigm (learning representation then classifier). It critiques standard decoupling methods for freezing the backbone in the second stage, which causes a significant loss of spatial and semantic information. BNS introduces a "Batch-No-Shift" module and an information-preservable objective, allowing the model to adjust the classifier while maintaining the integrity of the learned feature representation, thus bridging the gap between representation learning and classifier alignment.

- Focal-SAM (Focal Sharpness-Aware Minimization) (Li et al., 2025): Focal-SAM brings advanced optimization techniques into long-tailed recognition by adapting Sharpness-Aware Minimization (SAM). Recognizing that tail classes often converge to sharp, unstable minima in the loss landscape, Focal-SAM introduces a focal re-weighting strategy to the SAM objective. This forces the optimizer to prioritize finding flatter, more generalizable minima specifically for rare categories, thereby improving their robustness against test-time distribution shifts.

### D.2.2. FINE-TUNING FOUNDATION MODELS

For the setting of adapting Vision-Language Models (e.g., CLIP (Radford et al., 2021)) to long-tailed downstream tasks, we follow the standard protocol and compare with the following recent approaches:

- Decoder (VL-LTR) (Wang et al., 2024b): This method explores the integration of vision-language models into imbalanced learning. It appends a lightweight decoder structure to the fixed CLIP image encoder, which is trained to align visual features with their corresponding text embeddings. By explicitly leveraging the rich, pre-trained semantic priors from the text modality, the Decoder acts as a bridge that guides the visual classifier to recognize tail concepts that are visually under-represented but semantically rich.

- PECEL (Parameter-Efficient Complementary Expert Learning) (Ru et al., 2024): PECEL addresses the efficiency and adaptability trade-off by combining Mixture-of-Experts (MoE) with parameter-efficient fine-tuning. It creates a set of complementary experts (e.g., via LoRA or Adapters) that specialize in different data regimes (head vs. tail). A gating mechanism dynamically routes samples to the most appropriate expert, allowing the model to handle extreme class imbalances without the massive computational cost of full model fine-tuning.

- LIFT (Long-Tail Learning with Foundation Model) (Shi et al., 2024): LIFT provides a critical analysis of fine-tuning dynamics, revealing that "heavy" fine-tuning of foundation models on long-tailed data often leads to feature distortion and catastrophic forgetting of general knowledge. Instead, LIFT proposes a lightweight adaptation strategy that involves freezing most backbone parameters and tuning only a minimal set of bias terms or prompts. This preserves the robust open-world generalization capability of the pre-trained model while adapting it to the target distribution.

Applicability. It is important to emphasize that our proposed method operates orthogonally to these baselines. Whether it is an optimization-based approach like Focal-SAM, a representation learning method like BCL, or a foundation model adapter like LIFT, our method can be seamlessly integrated as a plug-and-play component to further boost performance across all these diverse settings.

### D.3. Additional Results

Tables 6 and 7 report split-wise results on CIFAR100-LT under IR=100 and IR=50, respectively. SRCD consistently improves the Tail split across all evaluated baselines in both imbalance settings, showing that it mainly benefits data-scarce classes. Under the more challenging IR=100 setting, SRCD improves Head, Medium, Tail, and overall accuracy for all baselines, with the most pronounced gains observed on the Tail split. Under IR=50, SRCD still improves Tail accuracy for every baseline; although CE+H2T shows a slight drop in overall accuracy, its Tail performance is still improved. Overall, these results further demonstrate that SRCD effectively enhances tail-class generalization, especially under severe long-tailed imbalance.

*Table 6.* Split-wise results on CIFAR100-LT under IR=100, reporting Top-1 accuracy (%) on Head/Med/Tail splits and All. +SRCD denotes integrating SRCD into each baseline; superscripts show absolute changes ($\uparrow/\downarrow$ for gain/drop).

| Method | Head | Med | Tail | All |
|---|---|---|---|---|
| GLMC | 63.39 | 60.95 | 34.63 | 53.91 |
| +SRCD | 63.88$_{\uparrow 0.49}$ | 62.37$_{\uparrow 1.42}$ | 36.90$_{\uparrow 2.27}$ | 55.26$_{\uparrow 1.35}$ |
| CE+H2T | 50.85 | 48.91 | 24.53 | 42.27 |
| +SRCD | 51.93$_{\uparrow 1.08}$ | 51.17$_{\uparrow 2.26}$ | 28.16$_{\uparrow 3.63}$ | 44.53$_{\uparrow 2.26}$ |
| BNS | 62.40 | 59.70 | 31.90 | 52.40 |
| +SRCD | 62.93$_{\uparrow 0.53}$ | 60.56$_{\uparrow 0.86}$ | 35.52$_{\uparrow 3.62}$ | 53.88$_{\uparrow 1.48}$ |
| Focal-SAM | 63.90 | 53.00 | 32.50 | 50.70 |
| +SRCD | 64.66$_{\uparrow 0.76}$ | 54.27$_{\uparrow 1.27}$ | 36.40$_{\uparrow 3.90}$ | 52.54$_{\uparrow 1.84}$ |

*Table 7.* Split-wise results on CIFAR100-LT under IR=50. reporting Top-1 accuracy (%) on Head/Med/Tail splits and All. +SRCD denotes integrating SRCD into each baseline; superscripts show absolute changes ($\uparrow/\downarrow$ for gain/drop).

| Method | Head | Med | Tail | All |
|---|---|---|---|---|
| GLMC | 64.73 | 62.10 | 38.19 | 58.87 |
| +SRCD | 65.51$_{\uparrow 0.78}$ | 63.53$_{\uparrow 1.43}$ | 41.86$_{\uparrow 3.67}$ | 60.43$_{\uparrow 1.56}$ |
| CE+H2T | 51.42 | 50.96 | 29.93 | 47.58 |
| +SRCD | 51.47$_{\uparrow 0.05}$ | 49.83$_{\downarrow 1.13}$ | 30.21$_{\uparrow 0.28}$ | 46.97$_{\downarrow 0.61}$ |
| BNS | 61.30 | 58.70 | 37.20 | 55.90 |
| +SRCD | 61.20$_{\downarrow 0.10}$ | 58.94$_{\uparrow 0.24}$ | 38.17$_{\uparrow 0.97}$ | 56.12$_{\uparrow 0.22}$ |
| Focal-SAM | 60.20 | 58.00 | 36.20 | 54.50 |
| +SRCD | 60.24$_{\uparrow 0.04}$ | 58.32$_{\uparrow 0.32}$ | 37.15$_{\uparrow 0.95}$ | 55.29$_{\uparrow 0.79}$ |

Table 8 further verifies the effectiveness of SRCD on Places-LT, a challenging long-tailed scene recognition benchmark. In the training-from-scratch setting, SRCD improves the overall accuracy of all evaluated baselines, with particularly clear gains on the Tail split. For example, SRCD improves the Tail accuracy of BCL, GLMC, GCL, and H2T by 1.32, 1.84, 0.58, and 2.96 percentage points, respectively, indicating that SRCD effectively improves tail-class recognition where shortcut reliance

is more severe. In the foundation-model fine-tuning setting, SRCD also brings consistent gains, improving the overall accuracy of PECEL, Decoder, and LIFT by 0.4, 0.7, and 1.0 points, respectively. Their Tail accuracy is also improved by 0.9, 1.5, and 2.3 points. Overall, the Places-LT results demonstrate that SRCD generalizes beyond object-centric long-tailed recognition and remains effective for long-tailed scene recognition.

*Table 8.* Results on Places-LT, reporting Top-1 accuracy (%) on Head/Med/Tail splits and All. The upper block is training from scratch, and the lower block is fine-tuning foundation models. +SRCD denotes integrating SRCD into each baseline; superscripts show absolute changes (↑/↓ for gain/drop).

| Method | Places-LT | | | |
|---|---|---|---|---|
| | Head | Med | Tail | All |
| **Training from scratch** | | | | |
| BCL | 41.98 | 42.36 | 34.91 | 40.77 |
| +SRCD | 41.85↓0.13 | 42.79↑0.43 | 36.23↑1.32 | 41.18↑0.41 |
| GLMC | 42.09 | 42.88 | 35.27 | 41.12 |
| +SRCD | 42.28↑0.19 | 43.05↑0.17 | 37.11↑1.84 | 41.62↑0.50 |
| GCL | 38.64 | 42.59 | 38.44 | 40.30 |
| +SRCD | 38.93↑0.29 | 42.74↑0.15 | 39.02↑0.58 | 40.65↑0.35 |
| H2T | 41.96 | 42.87 | 35.33 | 40.95 |
| +SRCD | 42.20↑0.24 | 42.64↓0.23 | 38.29↑2.96 | 41.64↑0.69 |
| **Fine-tuning foundation model** | | | | |
| Decoder | 46.3 | 46.8 | 45.8 | 46.4 |
| +SRCD | 46.5↑0.2 | 47.6↑0.8 | 47.3↑1.5 | 47.1↑0.7 |
| PECEL | 50.1 | 51.2 | 48.3 | 50.2 |
| +SRCD | 50.3↑0.2 | 51.4↑0.2 | 49.2↑0.9 | 50.6↑0.4 |
| LIFT | 51.3 | 52.2 | 50.5 | 51.5 |
| +SRCD | 51.6↑0.3 | 53.0↑0.8 | 52.8↑2.3 | 52.5↑1.0 |

## D.4. CAM Extraction Details for Fine-tuning Foundation Models

In the fine-tuning setting, we adapt a pre-trained vision(-language) foundation model (e.g., CLIP) to long-tailed recognition while explicitly supervising its spatial evidence via CAM. We use *CAM (weight-based evidence)* rather than Grad-CAM: the resulting map is computed from class weights and token/feature activations, without backpropagating gradients through the backbone for visualization. This choice makes the CAM branch lightweight, stable, and easy to integrate with parameter-efficient tuning (Adapters/LoRA), and it also matches the objective of constraining *where* the model looks under long-tailed bias.

### D.4.1. NOTATION AND GENERAL FORM

Let the image encoder output a spatial feature tensor. For CNN-style encoders, this is a feature map $A \in \mathbb{R}^{C \times H \times W}$. For ViT-style encoders, we treat patch tokens as a spatial grid $Z \in \mathbb{R}^{N \times D}$ with $N = H_p W_p$, and reshape it into $F \in \mathbb{R}^{D \times H_p \times W_p}$. Let the classifier (or linear probe) for class $c$ have weight vector $w_c$. The CAM for class $c$ is computed in a unified form:

$$M^c(\cdot) \;=\; \phi\Big(\big\langle w_c, \text{ spatial features}(\cdot)\big\rangle\Big), \tag{44}$$

where $\langle \cdot, \cdot \rangle$ denotes channel-wise inner product at each spatial location, and $\phi(\cdot)$ is a post-processing operator (e.g., ReLU + normalization + upsampling).

We always generate the CAM conditioned on the target class (ground-truth label) $t$, i.e., we use $M^t$ as the supervision target in the SRCD branch.

### D.4.2. ViT-based Encoders: Token-CAM (CAM on Patch Tokens)

Many CLIP variants use a ViT image encoder. In this case, the encoder produces a sequence of tokens after the last transformer block:

$$\tilde{Z} = [z_{\text{cls}}, z_1, \ldots, z_N] \in \mathbb{R}^{(N+1) \times D},$$

where $z_{\text{cls}}$ is the class token and $\{z_i\}_{i=1}^N$ are patch tokens. We build a spatial tensor from patch tokens by discarding $z_{\text{cls}}$ and reshaping:

$$F \in \mathbb{R}^{D \times H_p \times W_p}, \qquad F(:, u, v) = z_{i(u,v)}.$$

Then the Token-CAM for class $c$ is defined as:

$$M^c(u, v) \;=\; \phi\big(w_c^\top F(:, u, v)\big). \tag{45}$$

Practical choices that matter:

- Which token features to use. We use the output tokens after the final transformer block and its normalization (e.g., the last LayerNorm), because this is the feature space directly consumed by the classifier/projection head.

- Spatial resolution. For ViT-B/16, $(H_p, W_p) = (\frac{H}{16}, \frac{W}{16})$. The raw CAM is at patch resolution and is upsampled to the input image resolution using bilinear interpolation.

- Classifier weights. If we fine-tune a linear classifier on top of the frozen/pre-trained encoder, $w_c$ comes from this classifier head. If we use parameter-efficient modules (Adapters/LoRA) inside the encoder, $w_c$ is still the classifier head, while $F$ reflects the adapted token features.

**Post-processing $\phi$ for ViT Token-CAM.** We use a simple and stable $\phi$ to make CAM values comparable across samples:

$$\hat{M}^c = \max(M^c, 0) \quad \text{(ReLU)} \tag{46}$$

$$\tilde{M}^c = \frac{\hat{M}^c - \min(\hat{M}^c)}{\max(\hat{M}^c) - \min(\hat{M}^c) + \epsilon} \quad \text{(min-max norm)} \tag{47}$$

$$M_\uparrow^c = \text{Upsample}(\tilde{M}^c, \ H \times W) \quad \text{(bilinear)}. \tag{48}$$

This yields a bounded map $M_\uparrow^c \in [0, 1]^{H \times W}$ used by the SRCD loss.

### D.4.3. CNN-based Encoders: Standard CAM on Feature Maps

If the image encoder is CNN-style (e.g., ResNet-based CLIP), we extract the last convolutional feature map $A \in \mathbb{R}^{C \times H_f \times W_f}$ before global pooling. With a linear classifier weight $w_c \in \mathbb{R}^C$, the standard CAM is:

$$M^c(x, y) \;=\; \phi\Big(\sum_{k=1}^C w_{c,k}\, A_k(x, y)\Big). \tag{49}$$

The same $\phi$ (ReLU + min-max + bilinear upsampling) is applied to map it back to the image resolution.

### D.4.4. CLIP-specific Considerations (Vision-Language Encoders)

When using CLIP, there are two common classifier constructions:

- Linear classifier on visual features. We take the visual embedding $v$ (e.g., pooled $z_{\text{cls}}$ or pooled CNN feature) and train a classification head. CAM uses the head weights $w_c$ and spatial features from the visual backbone (Token-CAM for ViT or CAM for CNN).

- Text-prototype classifier (zero-shot style). If logits are computed by similarity between image embedding $v$ and text embeddings $\{t_c\}$, then a direct CAM needs an explicit linear weight $w_c$. In our fine-tuning setting, we use a learnable classification head for long-tailed adaptation, so $w_c$ is well-defined and CAM is straightforward.

This design ensures CAM is aligned with the downstream classifier used in evaluation.

### D.4.5. STOP-GRADIENT ON THE CAM BRANCH

In fine-tuning, especially with parameter-efficient tuning, we avoid destabilizing the main optimization by blocking gradients through the CAM construction path. Concretely, we compute CAM from detached spatial features:

$$F \leftarrow \text{detach}(F) \quad \text{or} \quad A \leftarrow \text{detach}(A),$$

and use it only for the SRCD loss computation. This prevents the CAM supervision from directly reshaping backbone features in a way that harms classification calibration, while still allowing SRCD to act as a consistent auxiliary signal through the designated parameters (e.g., SRCD-specific projection $\phi$, adapters, or a small CAM head if used).

### D.4.6. MULTI-CAM COLLECTION FOR SRCD TEACHER CONSTRUCTION (FINE-TUNING)

In the fine-tuning regime, we often exploit the stronger head-class priors of foundation models to build a robust teacher CAM. For each training sample, we form $K$ head-proxy CAMs (with $K$ fixed across all experiments; default $K = 3$):

1. obtain candidate proxy classes $\{c_1, \ldots, c_K\}$ (e.g., top-$K$ predictions among head classes or a predefined head-class pool);

2. compute their Token-CAM/CAM maps $\{M^{c_i}\}_{i=1}^K$ using Eq. (45) or Eq. (49);

3. aggregate them into a teacher CAM via an energy-based weighting scheme (details in Sec. 3.4).

This produces a teacher map that is typically more object-centric and less sensitive to long-tailed noise, which is crucial when tail classes have weak or diffuse evidence.

### D.4.7. IMPLEMENTATION DETAILS (HOOKS, SHAPES, AND EFFICIENCY)

We summarize practical details that are important for a correct and efficient implementation:

- Feature extraction point. For ViT, hook the output after the last transformer block (and its final LayerNorm if present) to get patch tokens. For CNN, hook the last convolutional stage before global pooling.

- Shape handling. For ViT, exclude the class token and reshape $N$ tokens into $(H_p, W_p)$. For non-square inputs, keep the true grid size derived from the patch embedding layer.

- Normalization stability. Use $\epsilon$ in min-max normalization to avoid division-by-zero on flat maps (common for early training).

- Upsampling. Bilinear upsampling is sufficient; we keep it deterministic for reproducibility.

- Memory/computation. CAM is computed from a single linear projection $w_c^\top F(:, u, v)$ at each location; the overhead is negligible compared to a forward pass. Blocking gradients further reduces autograd memory.

### D.4.8. DEFAULT HYPERPARAMETERS IN FINE-TUNING

Unless otherwise stated, we fix the SRCD loss weight to $\beta = 0.6$ and set the number of head-proxy CAMs to $K = 3$ for all fine-tuning experiments, keeping the rest of the training protocol (optimizer, learning rate schedule, epochs, batch size, and baseline-specific settings) unchanged for fair comparison.

### D.5. Comparison with a Center-Mask Prior

We further examine whether the gain of SRCD can be explained by a trivial geometric prior. Specifically, we construct a center-mask baseline on CIFAR100-LT under IR=100 using BNS as the baseline. This variant replaces the adaptive head-proxy teacher with a fixed mask centered in the image and uses it as the distillation target.

As shown in Table 9, the center-mask baseline performs worse than BNS, especially on the Tail split. This suggests that the gain of SRCD cannot be explained by simply encouraging attention toward the image center. A fixed center prior may suppress both object evidence and shortcut cues, and it can fail to focus on the actual object region when objects are

*Table 9.* Comparison with a center-mask baseline on CIFAR100-LT under IR=100. The center-mask baseline replaces the adaptive head-proxy teacher with a fixed centered mask.

| Method | Head | Med | Tail | All |
|---|---|---|---|---|
| BNS | 62.40 | 59.70 | 31.90 | 52.40 |
| BNS + Center Mask | 61.76$_{\downarrow 0.64}$ | 58.27$_{\downarrow 1.43}$ | 28.68$_{\downarrow 3.22}$ | 50.61$_{\downarrow 1.79}$ |
| BNS + SRCD | 62.93$_{\uparrow 0.53}$ | 60.56$_{\uparrow 0.86}$ | 35.52$_{\uparrow 3.62}$ | 53.88$_{\uparrow 1.48}$ |

off-center or have diverse layouts. In contrast, SRCD constructs an adaptive teacher from head-proxy CAMs and uses energy-based weighting to favor spatially coherent and concentrated evidence, which is crucial for improving tail-class recognition.

### D.6. Computational Overhead

**Overhead results.** We quantify the training overhead by comparing each baseline before and after integrating SRCD under an identical setup (same architecture, optimizer, learning-rate schedule, batch size, and total epochs). Table 10 reports the wall-clock time per iteration, total training time, throughput (images/sec). Under the current evaluated setting, adding SRCD incurs only a small increase in iteration time, while the overall training time remains close to the original baseline.

**Measurement protocol.** We measure (i) wall-clock time per iteration, (ii) total training time, and (iii) throughput (images/sec). All measurements are conducted on the same hardware/software environment and are averaged over multiple runs after a warm-up period.

*Table 10.* Training overhead on ImageNet with ResNet-50 (method GLMC). Reported Total time is the amortized wall-clock time over the full training run.

| Method | Time/iter (ms) | Total time (h) | Throughput (img/s) | $\Delta$ Time/iter |
|---|---|---|---|---|
| GLMC | 142 | 19.7 | 1842 | – |
| GLMC + SRCD | 153 | 22.6 | 1707 | +7.7% |

**Why the overhead is small.** The efficiency mainly comes from four design choices.

**Late-stage activation:** SRCD is enabled only in the late training stage (e.g., the last portion of epochs), so most iterations are identical to the baseline and the amortized overhead is reduced.

**Reusing forward features:** CAMs are computed from spatial representations already produced in the standard forward pass (e.g., the last CNN feature map or ViT patch tokens), requiring no extra backbone forward passes; CAM generation is a class-weighted linear projection plus a lightweight $\phi(\cdot)$ (ReLU/normalization/upsampling).

**No attribution backpropagation:** SRCD uses weight-based CAM with stop-gradient on the CAM branch, avoiding Grad-CAM-style gradient computation w.r.t. intermediate activations and thus avoiding an extra backward graph.

**Small constant factor ($K$):** teacher construction uses a small fixed number of head-proxy CAMs (default $K = 3$); each proxy CAM only adds a few linear projections on the reused feature map.

### D.7. Visualization of CAM Evidence

We visualize the spatial evidence during training to better understand how SRCD reshapes attention patterns under long-tailed bias. We provide two complementary views: (i) the diversity of head-proxy CAM candidates versus the tail ground-truth CAM, and (ii) the CAM quality change of the same backbone before and after integrating SRCD. These visualizations reveal two recurring phenomena: proxy CAMs are diverse and partially complementary, and SRCD consistently reduces shortcut-driven background attention while strengthening object-centric evidence.

**Head-proxy CAMs vs. tail ground-truth CAM.** Figure 6 presents multiple tail-class examples, where each row contains three head-proxy CAM candidates and the corresponding tail ground-truth CAM. Although each proxy is induced by a

head-biased hypothesis, high-quality candidates can still localize object-centric regions that overlap with the tail ground-truth evidence. In contrast, low-quality proxies frequently respond to scattered patterns or context/background cues, reflecting shortcut-driven correlations that are amplified by long-tailed training.

A key observation is that proxy CAMs are not identical: different proxies often emphasize different discriminative parts of the object (e.g., complementary regions), while their spurious responses are less consistent across proxies. This diversity is beneficial when aggregated properly. Our energy-based teacher construction explicitly leverages this property by favoring proxy CAMs that are spatially coherent and evidence-concentrated. As a result, the aggregated teacher tends to (i) combine complementary object evidence from multiple proxies into a more complete attention map, and (ii) suppress proxy-specific noise and background activations that do not persist across candidates. Empirically, this yields a teacher that is visually closer to the tail ground-truth CAM and is more stable across iterations, providing a robust supervision target for CAM alignment under severe class imbalance.

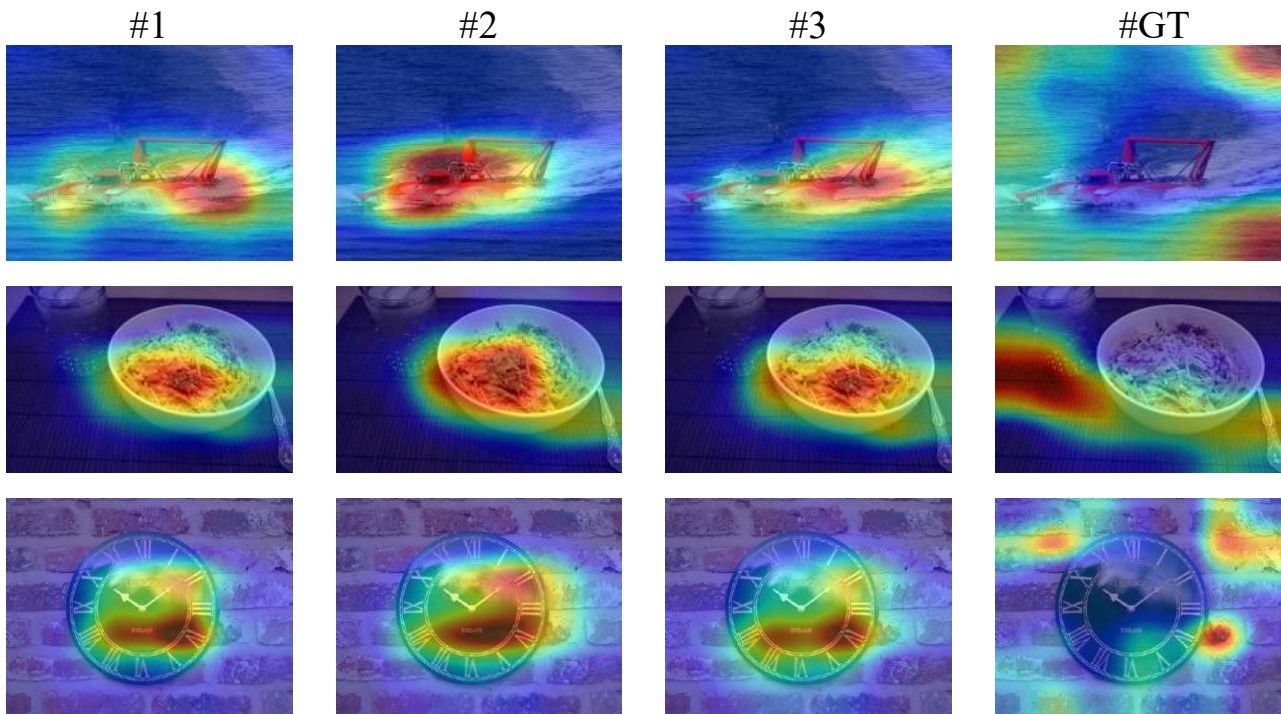

*Figure 6.* CAM visualizations on multiple tail-class examples. For each example (row), the first three columns show head-proxy CAM candidates, and the last column shows the tail ground-truth class CAM. High-quality proxies tend to capture object-centric evidence consistent with the ground truth, while low-quality ones highlight scattered or background-dominated regions.

**Effect of SRCD on CAM quality.** Figure 7 compares CAMs obtained from the same model without and with SRCD across multiple examples. Without SRCD, the model often exhibits shortcut-driven evidence: activations spread to background/context regions, and the attention mass is less spatially structured. This behavior is particularly pronounced for tail classes, where limited data encourages reliance on easy context cues rather than object evidence.

After integrating SRCD, CAMs become more concentrated on the true object with improved spatial coherence. In addition to sharper localization, we observe two qualitative improvements that are critical under long-tailed bias. First, SRCD reduces the dominance of single spurious cues by redistributing attention toward multiple object-relevant regions, resulting in more complete evidence coverage. Second, SRCD improves the consistency of attention across examples: the model relies on more stable object evidence instead of fluctuating background patterns. These observations are consistent with our design: SRCD uses a robust teacher (aggregated from diverse head-proxy candidates) to provide shortcut-resistant supervision, thereby pulling the student CAM away from background-biased modes and aligning it with object-centric evidence. Overall, the visualizations support that SRCD promotes more faithful evidence alignment and stronger shortcut resistance, especially for tail classes where spurious correlations are most harmful.

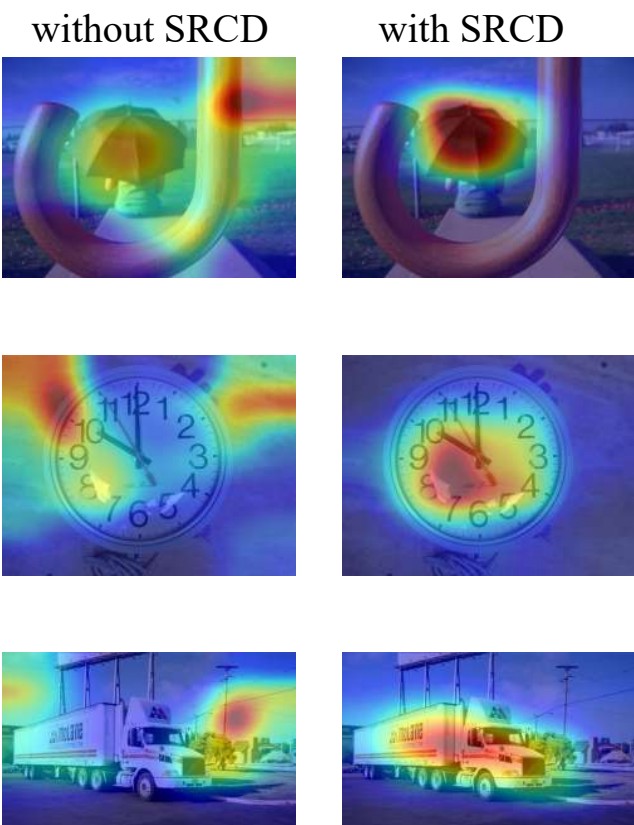

*Figure 7.* Effect of SRCD on CAM localization across multiple examples. Left: without SRCD. Right: with SRCD. SRCD promotes more object-centric and spatially coherent activations while reducing background/context responses, indicating stronger shortcut resistance.

*Table 11.* Mask-based shortcut analysis on the long-tailed VOC 2012 subset. FG/BG Ratio measures the CAM evidence mass inside the foreground mask relative to the background region. A larger ratio indicates weaker shortcut reliance.

| Method | FG/BG Ratio ↑ | Acc. ↑ |
|---|---|---|
| LIFT | 0.85 | 78.37 |
| LIFT + SRCD | 1.56 | 83.16 |

### D.8. Direct Evidence for Shortcut Suppression on VOC

Recognition accuracy alone may not fully reveal whether a method actually suppresses shortcut reliance. To provide more direct evidence, we conduct a mask-based analysis on Pascal VOC 2012, whose semantic segmentation masks enable foreground-background evaluation. Since the original VOC 2012 dataset is multi-label, we curate a single-label subset and reorganize it into a long-tailed distribution consistent with our setting, with an imbalance ratio of 80.

We use LIFT as the baseline and report the Foreground-to-Background Ratio (FG/BG Ratio), which measures how CAM attention is distributed inside versus outside the segmentation mask:

$$\text{FG/BG} = \frac{\sum_{u \in \Omega_{\text{fg}}} \bar{M}(x)_u}{\sum_{u \in \Omega_{\text{bg}}} \bar{M}(x)_u + \epsilon},$$

where $\Omega_{\text{fg}}$ and $\Omega_{\text{bg}}$ denote the foreground and background regions given by the segmentation mask, respectively. A larger FG/BG ratio indicates that the model assigns more evidence to object regions and relies less on background shortcuts.

As shown in Table 11, SRCD substantially increases the FG/BG Ratio from 0.85 to 1.56, while also improving accuracy from 78.37 to 83.16. This provides direct evidence that SRCD shifts attention toward object regions and away from background or shortcut regions, rather than merely improving recognition accuracy.

We further provide VOC CAM visualizations in Figure 8. Compared with the baseline, SRCD produces CAMs that are more concentrated on foreground object regions and less dominated by background context. These qualitative results are consistent with the mask-based FG/BG analysis and further support that SRCD alleviates shortcut-dominated attention in practice.

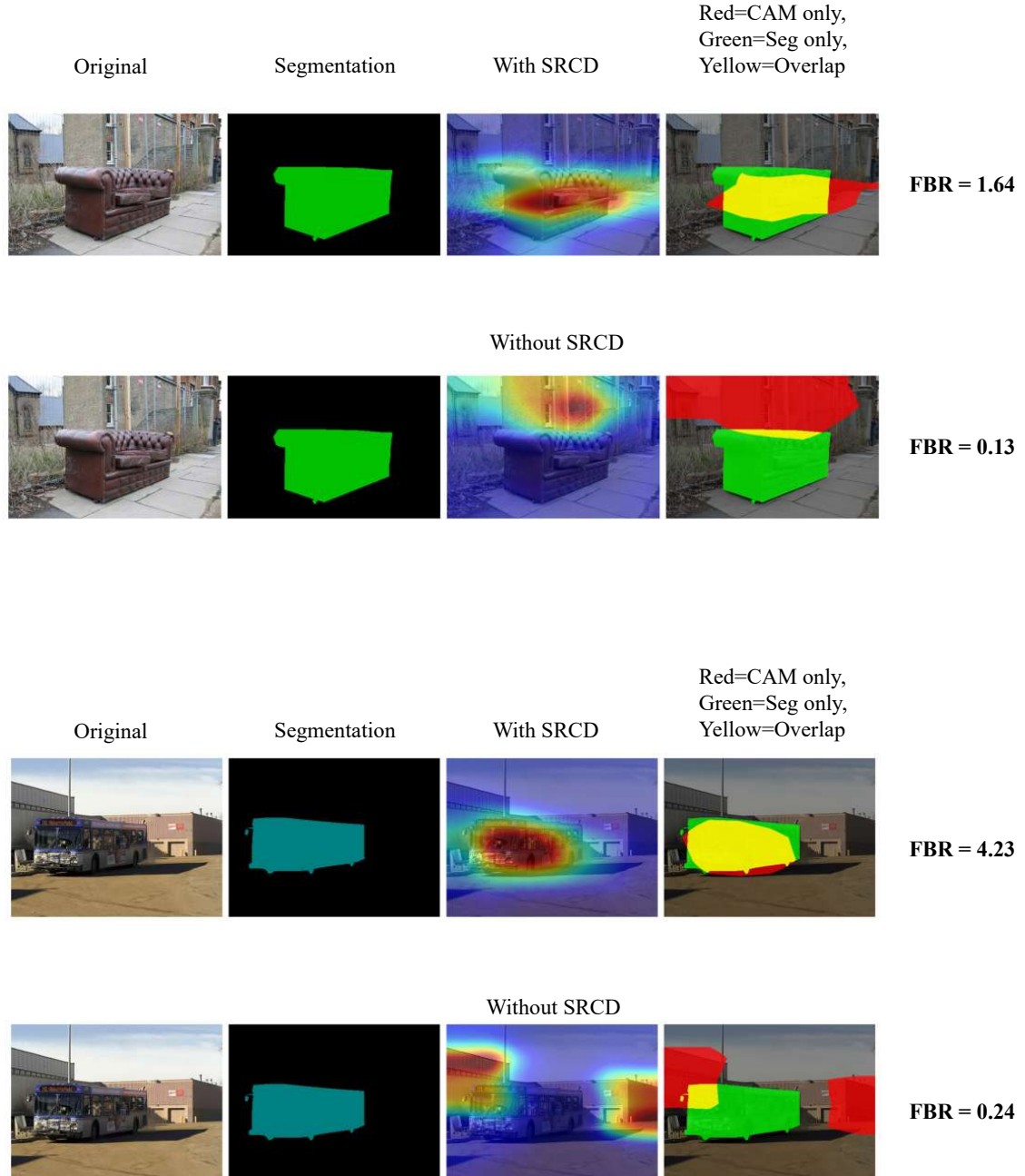

*Figure 8.* CAM visualizations on the long-tailed VOC 2012 subset. Compared with the baseline, SRCD shifts attention from background or shortcut regions toward foreground object regions, which is consistent with the mask-based FG/BG analysis.

# E. Algorithm

---

**Algorithm 1** Shortcut-Resistant CAM Distillation (SRCD)

---

**Require:** Mini-batch $\mathcal{B}$, model $f_\theta$, head classes $\mathcal{Y}_{\text{head}}$, tail classes $\mathcal{Y}_{\text{tail}}$, proxy number $K$, temperature $\tau$, weight $\beta$.

1: Forward $f_\theta$ to obtain logits $z_i$ and feature maps $A_i$.
2: Compute classification loss $\mathcal{L}_{\text{cls}}$.
3: Initialize $\mathcal{L}_{\text{SRCD}} \leftarrow 0$.
4: **for** each $(x_i, y_i) \in \mathcal{B}$ with $y_i \in \mathcal{Y}_{\text{tail}}$ **do**
5:     Select top-$K$ predicted head classes:
$$\mathcal{H}(x_i) = \text{TopK}_{c \in \mathcal{Y}_{\text{head}}} \, p_i(c).$$

6:     Compute the student tail CAM without stop-gradient:
$$\bar{M}_{y_i}(x_i) = \text{Norm}(\text{CAM}(A_i, y_i)).$$

7:     Compute detached head-proxy CAMs:
$$\bar{M}_c^{\text{sg}}(x_i) = \text{sg}[\text{Norm}(\text{CAM}(A_i, c))], \quad c \in \mathcal{H}(x_i).$$

8:     Compute proxy scores and weights:
$$s_c = -\mathcal{E}_{\text{h}}(\bar{M}_c^{\text{sg}}) + \mathcal{E}_{\text{n}}(\bar{M}_c^{\text{sg}}), \quad w_c = \frac{\exp(s_c/\tau)}{\sum_{c' \in \mathcal{H}(x_i)} \exp(s_{c'}/\tau)}.$$

9:     Construct the detached teacher CAM:
$$\bar{M}_T(x_i) = \text{sg}\left[ \sum_{c \in \mathcal{H}(x_i)} w_c \bar{M}_c^{\text{sg}}(x_i) \right].$$

10:    Accumulate SRCD loss:
$$\mathcal{L}_{\text{SRCD}} \leftarrow \mathcal{L}_{\text{SRCD}} + \gamma_{y_i} \left\| \bar{M}_{y_i}(x_i) - \bar{M}_T(x_i) \right\|_2^2.$$

11: **end for**
12: Optimize:
$$\mathcal{L} = \mathcal{L}_{\text{cls}} + \beta \mathcal{L}_{\text{SRCD}}.$$

13: Update $\theta$ by back-propagating $\mathcal{L}$.

---

