# OpenReview forum: "Shortcut-Resistant CAM Distillation for Long-Tailed Recognition"
_ICML.cc/2026/Conference — ICML 2026 regular_

### Official Review · Reviewer_PAPX · 2026-03-12

**Soundness:** 3
**Presentation:** 3
**Significance:** 3
**Originality:** 3
**Overall Recommendation:** 5
**Confidence:** 3

**Summary:**

This work addresses the common long-tailed distribution problem in real-world visual data from a new perspective of shortcut learning, and finds the obvious attention difference between head and tail classes in long-tailed recognition. Specifically, because tail classes have few samples and similar backgrounds, the model is likely to depend on non-semantic false clues such as background and texture for shortcut learning. In contrast, the abundant and various samples of head classes can guide the model to learn stable object-focused features. Based on this key finding, the paper proposes the Shortcut-Resistant CAM Distillation (SRCD) framework. It collects an anti-shortcut teacher signal from candidate CAMs of head classes with an energy model weighting method, and then passes this signal to the CAMs of tail classes, thus reducing the model reliance on shortcuts in tail classes. The paper also provides a theoretical analysis for SRCD, which measures how much the model relies on shortcuts in the CAM space and proves that the framework works well in reducing shortcuts for tail classes. Lots of experiments on many common long-tailed recognition datasets show that this plug-and-play framework can steadily boost the performance of various baseline methods.

**Compliance With Llm Reviewing Policy:**

Affirmed.

**Final Justification:**

Thank you for your detailed response and extensive experimental validations. Most of my previous questions have been properly addressed. The research idea and perspective of this paper are valuable, which can provide important research inspiration for relevant fields and facilitate the exploration of better technical solutions. Therefore, I decide to raise my review score for this paper.

**Key Questions For Authors:**

1. The division of object regions and shortcut regions in this paper is only conceptual. Have the authors attempted to quantitatively verify the core conclusion that SRCD significantly reduces the attention mass in shortcut regions using object segmentation annotations? How to prove that the method truly suppresses shortcut dependence, rather than merely optimizing classification accuracy?
2. Visualization results show that accurate attention maps are mainly concentrated in the central region. Can you directly constrain the attention region to the image center without complex designs such as energy weighting? If so, can this simple strategy also bring performance improvements?
3. When selecting head-class proxies, the model may select incorrect classes if it is affected by severe biases. Although energy weighting is adopted in the method, can the authors guarantee that the aggregated teacher CAM will not be dominated by incorrect proxies under extreme bias conditions?

**Limitations:**

yes

**Strengths And Weaknesses:**

Strengths：
1. Novel research perspective: Previous studies on long-tailed recognition mainly focused on alleviating the scarcity of tail-class data through data rebalancing, classifier calibration and other strategies, while ignoring the intrinsic cause of the performance collapse of tail classes. This work conducts an investigation into the inherent correlation between the performance gap of head and tail classes, and for the first time systematically links shortcut learning to the head-tail performance gap in long-tailed recognition. It reveals the core mechanism that the scarcity of tail-class data and contextual homogeneity may jointly contribute to shortcut dependence.
2. Comprehensive theoretical system: The work formulates a quantitative index for shortcut dependence in the CAM space, transforming the abstract problem of shortcut learning into a quantifiable and optimizable mathematical problem. It proves the inherent rationality of energy-weighted teacher aggregation via the entropy-regularized utility maximization theorem, and verifies the monotonic contraction characteristic of L2 distillation on shortcut attention through gradient flow analysis, thus rigorously guaranteeing the method’s effectiveness from a theoretical perspective.
3. Sufficient comparative experiments: The work completes comprehensive validations on four mainstream long-tailed recognition benchmarks, which demonstrates the universality of the proposed method. It verifies the effectiveness of core modules through ablation experiments and conducts qualitative analysis combined with CAM visualization, where the quantitative and qualitative conclusions mutually corroborate each other.


Weaknesses：
1. Question on innovation: The innovation of this work mainly lies in theoretically and empirically analyzing the impact of shortcut learning on long-tailed recognition, and designing a CAM alignment distillation strategy to suppress shortcut learning during model training. However, it fails to clearly demonstrate the unique design of the proposed method in the long-tailed scenario. In addition, the basic paradigm of CAM distillation is not pioneered in this work, which limits the originality of the contribution.
2. Incomplete experimental validation: The core harm of shortcut learning is the collapse of out-of-distribution (OOD) generalization ability. Although the proposed method claims to alleviate shortcut learning, the performance improvement on recognition datasets alone cannot prove that the method truly mitigates shortcut learning. It may only alleviate the overfitting of long-tailed samples.
3. Insufficient verification on design: The proposed energy-model aggregation mechanism involves two indicators (coherence and concentration), resulting in a relatively complex design. Nevertheless, visualization results show that effective attention maps tend to concentrate on the central region of objects. It remains unclear whether simple geometric priors (e.g., focusing on the image center) can achieve similar effects. The lack of comparison with simple baselines makes it difficult to measure the marginal gain of the complex design in this work.

---

> ### Author Rebuttal · Authors · 2026-03-30
>
> We sincerely thank the reviewer for the thoughtful comments and constructive questions. We respond to them below.
>
> ## W1: Long-Tailed-Specific Design and Innovation
>
> Our work is motivated by the long-tailed setting, where the key challenge is not only class imbalance itself, but also the asymmetric shortcut vulnerability it induces: tail classes, under scarce and biased supervision, are more likely to develop shortcut-dominated attention, while head classes tend to learn relatively more stable object-focused evidence. Based on this long-tailed-specific observation, SRCD is designed to use more reliable head-side explanations to regularize tail-class attention during training. In this sense, the novelty of our method does not lie in CAM distillation alone, but in how the distillation is specifically constructed around the head–tail structure: the teacher is built from head-class proxies, the correction is applied specifically to tail classes, and the regularization strength is scaled by class rarity. Therefore, SRCD is not a generic CAM distillation strategy directly applied to long-tailed data, but a long-tailed-motivated mechanism for correcting shortcut-dominated tail attention.
>
>
>
> ## W2: Direct evidence for shortcut suppression
>
> We agree that recognition accuracy alone is insufficient to fully establish shortcut suppression. While Sec. 3.3 already provides initial evidence through CAM similarity and qualitative results, we further aim to provide more direct support. This is difficult on standard long-tailed benchmarks, since they do not provide pixel-level annotations for explicitly measuring whether attention shifts from background regions to foreground objects. To address this, we additionally conduct a quantitative analysis on Pascal VOC 2012, whose semantic segmentation masks enable direct mask-based evaluation. Since the original VOC 2012 dataset is multi-label, we curate a single-label subset and reorganize it into a long-tailed distribution consistent with our setting, with an imbalance ratio of 80.
>
> Using LIFT as the baseline, we report a Foreground-to-Background Ratio (FG/BG Ratio), which measures how CAM attention is distributed inside versus outside the segmentation mask; a larger ratio indicates weaker shortcut reliance.
>
> | Method      | FG/BG Ratio ↑ |  Acc. ↑   |
> | -| :-: | :-: |
> | LIFT   |     0.85      |   78.37   |
> | LIFT + SRCD |   **1.56**    | **83.16** |
>
> These results provide more direct evidence that SRCD shifts attention toward object regions and away from background/shortcut regions, rather than merely improving accuracy. In addition, the CAM visualizations in the Appendix consistently support that SRCD alleviates shortcut-dominated attention in practice.
>
> ## W3: Necessity of the energy-based design
>
> We agree that simple baselines are important for evaluating the necessity of our design. To test whether the gain can be explained by a trivial geometric prior, we add a center-mask baseline on CIFAR100-LT (IR=100) using BNS as the baseline. Specifically, we replace the head-proxy teacher with a fixed mask centered in the image and use it as the distillation target.
>
> | Method | Head |       Med |      Tail |       All |
> | :-| -: | -: | -: | -: |
> | BNS |     62.40 |     59.70 |     31.90 |     52.40 |
> | BNS + Center Mask |     61.76 |     58.27 |     28.68 |     50.61 |
> | BNS + SRCD        | **62.93** | **60.56** | **35.52** | **53.88** |
>
> The center-mask baseline performs worse than BNS, while SRCD consistently improves all splits, especially the tail split. This suggests that the gain of SRCD cannot be explained by simply encouraging attention toward the image center. A fixed center prior may suppress both object evidence and shortcut cues, yet still fail to focus on the actual object region, especially when objects are off-center or have diverse layouts. In contrast, SRCD benefits from adaptive object-centric evidence transfer.
>
> ## Q1: Verification with segmentation annotations
>
> Related to W2.
>
> ## Q2: Can a center prior explain the gain?
>
> Related to W3.
>
> ## Q3: Robustness to incorrect head proxies
> Our method is motivated by the head–tail asymmetry in long-tailed learning. Head classes have richer and more diverse supervision, so they usually learn more stable and object-focused evidence. Tail classes have scarce and biased data, so they are more easily dominated by shortcut cues. This is why SRCD uses relatively more reliable head-side explanations to regularize noisier tail attention. That said, we do not claim a strict guarantee that incorrect proxies will never dominate the aggregated teacher under extreme bias. Instead, our method is designed to mitigate this risk as much as possible. Its practical basis is that head-side CAMs are often more stable and less shortcut-prone than tail-side CAMs in the long-tailed setting. We will clarify this assumption and its limitation more explicitly in the revision.
>
> Overall, we thank the reviewer again for the constructive comments.

---

> > ### Author Rebuttal · Reviewer_PAPX · 2026-04-06
> >
> > Thank you for your detailed response. The experimental metrics on the Pascal VOC dataset have addressed most of my doubts, and it would be even better if some visual illustrations could be added. As for the deficiencies in the paper’s presentation raised by the reviewer above, I think there is still room for improvement. For instance, in Figure 3, the paper mentions multiple head proxies, but the figure only displays three CAMs of the same image.
> > Mitigating shortcut learning can advance the development of other fields, and I believe this paper will offer us valuable relevant insights.

---

> > > ### Author Response · Authors · 2026-04-07
> > >
> > > Thank you for your positive and thoughtful comments. We are encouraged that the Pascal VOC results have addressed most of your concerns.
> > >
> > > ## Additional visual illustrations
> > >
> > > We agree that visual illustrations would further improve the paper. To make the shortcut-mitigation effect more intuitive, we will add more qualitative examples in the revision. We have already prepared many visual illustrations [here](https://anonymous.4open.science/r/Figure_2-C3E5/voc_1.pdf).
> > >
> > > ## Clearer presentation
> > >
> > > - We will clarify this point more clearly in the revision in response to Reviewer 7UZy. Specifically, **we do not expect different head proxies to provide perfectly complementary part coverage or to fully match the target object**. In fact, even for head classes with abundant samples, this is already difficult to achieve. So it is not realistic to require multiple proxies on tail examples to cover different object parts while also strictly matching the target object. This is not the goal of our method. Instead, we study long-tailed recognition from a different perspective. Most existing methods mainly focus on representation bias or classifier bias caused by class imbalance, while we further observe that tail-class CAMs are also more easily affected by shortcut cues. We believe this is an important issue in long-tailed recognition, but it has not been sufficiently studied or emphasized in prior work. SRCD is proposed for this reason. Its goal is not to pursue strict part-level decomposition, but to build a teacher CAM that is more stable and less affected by shortcuts than the original tail CAM. Even if this teacher does not perfectly cover every object part, it can still provide useful supervision as long as it is less shortcut-prone than the original tail attention. We will make this goal and boundary clearer in the revised paper.
> > > - We also appreciate your comment on Figure 3. Since the figure only shows three CAMs from the same image. To make this part more intuitive, we have added more descriptions to Figure 3 so that the roles of multiple head proxies and the teacher aggregation process are easier to follow. The updated version is available [here](https://anonymous.4open.science/r/Figure_2-C3E5/framework_new.pdf).
> > >
> > >
> > >
> > > ## Clarification on the Comment from the Newly Added Reviewer Eiig
> > >
> > > Reviewer Eiig was added after the rebuttal period. Due to the rebuttal constraints, we were unable to respond directly during the discussion phase. We therefore clarify it here. The reviewer cites ResLT to argue that our novelty is limited because prior work has incorporated residual learning into long-tailed recognition. However, this comparison is not valid. ResLT is essentially a multi-branch classification method, where the main branch makes a base prediction and the additional branches compensate for the medium- and tail-class predictions. Its residual learning is therefore a classifier-level output fusion design. In contrast, our work studies shortcut learning as a distinct failure mode in long-tailed recognition, with a particular focus on the biased shortcut cues learned by tail classes during training. The two works are fundamentally different in both research focus and technical design, and should not be treated as the same type of method.
> > >
> > >
> > >
> > > We will incorporate these visual materials and clarifications into the revised version to further improve the presentation.
> > >
> > > Thank you again for your thoughtful and constructive feedback!

---

### Official Review · Reviewer_7UZy · 2026-03-12

**Soundness:** 2
**Presentation:** 3
**Significance:** 3
**Originality:** 3
**Overall Recommendation:** 4
**Confidence:** 4

**Summary:**

The paper addresses the long-tailed recognition problem by the lens of shortcut learning, and proposes Shortcut-Resistant CAM Distillation (SRCD) to optimize tail class recognition performance. SRCD first generate CAMs from a set of predicted head classes for a given tail sample, and then aggregates the proxy CAMs as a teacher to suppress shortcut reliance for CAM of the tail class. Experimental results demonstrate consistent improvements when SRCD is added to various long-tailed learning baselines.

**Compliance With Llm Reviewing Policy:**

Affirmed.

**Final Justification:**

The proposed method proposes an interesting view for long-tailed recognition, and the authors have addressed most of my concerns. I would raise the score to 4.

**Key Questions For Authors:**

1. Could the authors provide an empirical analysis of the results when the top-$K$ predicted head classes are semantically unrelated to the tail class?
2. Since CAMs inherently focus on local discriminative patches rather than entire objects, how does SRCD guarantee that its distillation transfers holistic object-centric knowledge rather than this partial-object bias?
3. In fine-grained datasets such as iNaturalist, would the performance drop on irregular object classes?

**Limitations:**

The proposed method relies on CAMs and rigid energy metrics, which tends to inherit partial-object biases while discarding valid contextual evidence for irregular objects.

**Strengths And Weaknesses:**

Strengths
1. Treating the long-tailed recognition problem through the intersection of shortcut learning and spatial attribution (CAMs) is a novel perspective.
2. Using Laplacian smoothness and Hoyer sparsity to define an energy-based weighting scheme provides a mathematically grounded way to select and aggregate the most object-centric proxy CAMs without requiring external bounding box annotations. Moreover, it's architecture-agnostic.

Weaknesses
1. The core assumption of the paper relies on that head-proxy CAMs can provide accurate object-centric localization for tail class images. However, if the two classes are completely unrelated, it' hard to guarantee that mismatched head-proxy classifiers reliably highlight the correct semantic object regions for tail class images.
2. The proposed method relies heavily on the quality of CAMs while CAM does not always align with object-centric semantics, which will make the distillation target noisy.
3. Laplacian smoothness and Hoyer sparsity tend to penalize diffuse or fragmented attention, which is inappropriate for irregular objects such as insects.

---

> ### Author Rebuttal · Authors · 2026-03-30
>
> We are grateful for the reviewer’s constructive comments and respond to them below.
>
> ## W1: Semantic Mismatch of Head Proxies
>
> We would like to clarify that our method does not rely on strict semantic matching between the head proxy and the tail class. The key intuition is that object-centric CAM localization is largely a shared model capability, but under long-tailed training, head classes benefit from richer and more diverse supervision and therefore tend to learn more reliable CAMs, whereas tail classes are more prone to shortcut-dominated CAMs due to scarce and biased data. Accordingly, SRCD uses relatively more reliable head-class explanations to regularize noisier tail-class attention, preventing the model from further reinforcing spurious localization patterns during training. In this sense, the key is not exact semantic correspondence, but whether the proxy CAM provides a cleaner spatial prior than the original tail CAM.
>
> To directly examine the reviewer’s concern, we conduct an additional analysis on iNaturalist using taxonomy labels (e.g., Actinopterygii, Amphibia, Animalia, Arachnida, Aves, Insecta, Mammalia, Plantae) to identify semantically unrelated head proxies. With BCL as the baseline, 32.13% of tail samples fall into this mismatched subset. Importantly, SRCD still improves the accuracy on these samples from 67.25 to 69.88. This result suggests that SRCD does not strictly rely on semantic matching between the head proxy and the tail target. Even when the selected proxy is taxonomically unrelated, its CAM can still serve as a better spatial prior as long as it is less shortcut-prone than the original tail CAM.
>
> | Method | Acc. |
> | :-| :-: |
> | BCL|   67.25   |
> | BCL + SRCD | **69.88** |
>
> ## W2: Imperfect CAM Supervision
>
> We agree that CAMs are not always perfectly object-centric. Rather than assuming perfect CAMs, SRCD is explicitly designed to handle this imperfection in the long-tailed setting. Since head-side CAMs are often relatively more reliable than shortcut-dominated tail CAMs, SRCD uses them to regularize and correct noisy tail-class attention, preventing the model from further reinforcing spurious localization patterns during training. This is further stabilized by proxy aggregation and late-stage activation. We will clarify this point in the paper.
>
> ## W3: Limited Suitability for Irregular Objects
>
> This concern mainly applies to a single CAM, whereas our teacher is an energy-weighted aggregation of multiple head-proxy CAMs rather than a single rigid attention map. The smoothness and sparsity terms only score each candidate CAM individually; after aggregation, different proxies can attend to different object parts, which is exactly why the method remains suitable for irregular objects such as insects. In our [linked visualization](https://anonymous.4open.science/r/Figure_2-C3E5/Example_of_Irregular_Objects.pdf) with Stagmomantis limbata (a species of mantis), different head-proxy CAMs focus on different regions, and their weighted combination yields a much clearer object-centric teacher CAM. We will clarify this point in the paper.
>
> ## Q1: Empirical Analysis Under Proxy–Target Semantic Mismatch
>
> Related to W1.
>
>
>
> ## Q2: From Local Discriminative Patches to Holistic Knowledge
>
> While a single CAM often focuses on local discriminative patches, SRCD does not distill from a single CAM. Instead, we generate CAMs from multiple head proxies and construct the teacher through energy-weighted aggregation, allowing different proxies to capture different object parts. As a result, the distilled teacher is not merely the local bias of one CAM, but a more complete and stable object-related spatial prior.
>
> ## Q3: Performance on Irregular Objects
>
> To examine this concern, we conduct an analysis on iNaturalist using BCL as the baseline, focusing on tail classes under the Insecta category, which serves as a representative group of irregular objects. We find that applying SRCD improves the accuracy on this subset from 67.13 to 69.50. This result suggests that SRCD does not deteriorate performance on irregular object classes. A plausible reason is that the smoothness/sparsity cues are used only to score individual head-proxy CAM candidates, while the final teacher is an aggregation of multiple head-proxy CAMs. As discussed earlier, different proxies can attend to different object parts, and their combination can still form a clearer object-centric spatial prior even for structurally irregular objects such as insects. We will add this analysis to clarify the behavior of SRCD on fine-grained irregular categories.
>
> |     Method       |   Acc.    |
> | :--------- | :-------: |
> | BCL        |   67.13   |
> | BCL + SRCD | **69.50** |
>
>
>
> Overall, we thank the reviewer again for these valuable suggestions.

---

> > ### Author Rebuttal · Reviewer_7UZy · 2026-04-03
> >
> > Thank you for your detailed response. Some of concerns are still not clearly addressed. I have some follow-up questions.
> >
> > Although SRCD generates CAMs from multiple head proxies, I find no explicit algorithmic mechanism in SRCD to guarantee that different proxies attend to different object parts. The method independently scores each CAM (${\huge\varepsilon}_h$ and ${\huge\varepsilon}_n$) and performs a weighted sum. There is no spatial diversity constraint or orthogonality penalty to prevent multiple top-$K$ proxies from collapsing into the exact same discriminative local patch. Although the supplementary experiments provide some empirical evidence of its effectiveness, the underlying mechanism regarding the partial-object bias remains theoretically unresolved.

---

> > > ### Author Response · Authors · 2026-04-03
> > >
> > > ## On the Lack of Explicit Part-Level Diversity Across Proxies
> > >
> > > Thank you for this insightful comment. We appreciate the opportunity to clarify this point. **We do not expect different head proxies to provide perfectly complementary part coverage or to fully match the target object**. In fact, even for head classes with abundant samples, this is already difficult to achieve. So it is not very realistic to require multiple proxies on tail examples to cover different object parts while also strictly matching the target object. Enforcing such a property explicitly would likely require a larger top-K candidate set, finer pixel-level comparisons across attention maps, and additional diversity or structural constraints, which would also bring higher computational cost. But even with this added complexity, it would still be difficult to verify whether the resulting parts are truly correct and complementary.
> > >
> > > We study long-tailed recognition from a different perspective. Most existing methods mainly focus on representation bias or classifier bias caused by class imbalance. We further observe that tail-class CAMs are also more easily affected by shortcut cues. We believe this is an important issue in long-tailed recognition, but it has not been sufficiently studied or emphasized in prior work. SRCD is proposed for this reason. Its goal is not to pursue strict part-level decomposition, but to build a teacher that is more stable and less affected by shortcuts than the original tail CAM. Even if this teacher does not perfectly cover every object part, it can still be useful as long as it is less shortcut-prone than the original tail attention. This may still hold for irregular objects such as insects, and our earlier experimental results also support this point. Thank you for your constructive comments. We will make this goal and boundary clearer in the revised paper.
> > >
> > >
> > > ## Clarification on the Comment from the Newly Added Reviewer Eiig
> > >
> > > Reviewer Eiig was added after the rebuttal period. Due to the rebuttal constraints, we were unable to respond directly during the discussion phase. We therefore clarify it here. The reviewer cites ResLT to argue that our novelty is limited because prior work has incorporated residual learning into long-tailed recognition. However, this comparison is not valid. ResLT is essentially a multi-branch classification method, where the main branch makes a base prediction and the additional branches compensate for the medium- and tail-class predictions. Its residual learning is therefore a classifier-level output fusion design. In contrast, our work studies shortcut learning as a distinct failure mode in long-tailed recognition. SRCD operates in the CAM space, using head-class proxy CAMs to build a more reliable teacher and distilling it to the tail-class CAM to mitigate shortcut reliance. Thus, SRCD is essentially an evidence-level method. The two works are fundamentally different in motivation, research focus, and technical design, and therefore should not be viewed as the same type of method.
> > >
> > > We will incorporate these clarifications into the revised paper to make the presentation clearer.
> > >
> > > Thank you again for your thoughtful and constructive feedback!

---

### Official Review · Reviewer_XbCh · 2026-03-19

**Soundness:** 3
**Presentation:** 3
**Significance:** 3
**Originality:** 3
**Overall Recommendation:** 4
**Confidence:** 4

**Summary:**

To improve the long-tailed learning especially the tail classes, this paper revisit the problem from the angle of shortcut learning and observe that the shortcut learning is amplified for tail classes since we have limited examples sharing similar contexts. Based on this observation, this paper proposes a shortcut-resistant CAM distillation by transferring object-focused explanations from top likely head classes to tail classes. During the transfer, only those with spatially connected and concentrated CAMs would be given high energy. Theoretically analysis is also provided to prove that the proposed method can help suppress tail shortcuts. Experiments on four benchmark long tail datasets show that the proposed method can be plugged into existing long tailed methods (training from scratch or fine-tuning) and help improve the performance. The ablation study also shows the contribution of the energy-based components.

**Compliance With Llm Reviewing Policy:**

Affirmed.

**Final Justification:**

Thanks for addressing my concerns. I would love to keep my positive score.

**Key Questions For Authors:**

Please check the above weak points.

**Limitations:**

Yes.

**Strengths And Weaknesses:**

Strong points:
1. Based on the revisit of the long-tailed problem from the angle of shortcut learning, this paper proposes to transfer object-focused CAMs from top similar head classes to tail classes is reasonable and novel.
2. The energy-driven idea to transfer only spatially connected and concentrated CAMs is also interesting and sound.
3. This paper also provide a theoretical analysis to prove the proposed method can suppress the tail shortcut, which theoretically demonstrates the proposed method.
4. Performance improvement can be observed on four long tail benchmarks by plugging the proposed method to existing approaches.

This paper can be further improved by addressing the following weak points:
1. Following the common practice, including the benchmark long tailed data of "places" will help demonstrate the proposed method further.
2. Considering that the proposed method is mainly targeted to help improve the tail classes, the specific performance on the tail classes can be emphasized not just for two datasets but also for the remaining datasets and the ablation study.
3. The writing can be improved. For example, the definitions of head, med, tail should be made clear and consistent through the paper. The reference of "Table 5 (b)" is hard to follow for "Comparison with Averaging Strategy".

---

> ### Author Rebuttal · Authors · 2026-03-30
>
> We sincerely thank the reviewer for the constructive suggestions. We agree that the paper can be further strengthened by adding more benchmarks, reporting tail-class results more comprehensively, and improving clarity in presentation.
>
> ## 1. On including Places-LT
>
> Thank you for this valuable suggestion. We agree that including Places-LT would further strengthen the empirical evaluation. Following your suggestion, we conducted experiments on Places-LT.
>
> We choose Decoder and LIFT for this study because they are both strong and representative baselines in our foundation-model fine-tuning setting, and they were already used in our main experiments. Specifically, Decoder explicitly leverages vision-language semantic priors through a lightweight decoding module, making it a representative CLIP-based adaptation baseline for tail recognition. LIFT emphasizes lightweight adaptation that preserves the pre-trained model’s general knowledge while reducing feature distortion during long-tailed fine-tuning, making it a strong baseline for evaluating whether SRCD brings complementary gains for tail classes. The results are shown below:
>
> | Method         | Head |  Med | Tail |  All |
> | -| ---: | ---: | ---: | ---: |
> | Decoder        | 46.3 | 46.8 | 45.8 | 46.4 |
> | Decoder + SRCD | 46.5 | 47.6 | 47.3 | 47.1 |
> | LIFT           | 51.3 | 52.2 | 50.5 | 51.5 |
> | LIFT + SRCD    | 51.6 | 53.0 | 52.8 | 52.5 |
>
> These results further support our claim that SRCD consistently improves tail performance while maintaining strong overall accuracy. We will include the Places-LT results, together with the full set of method comparisons, in the revised version.
>
>
> ## 2. On emphasizing tail-class performance
>
> We agree with the reviewer that, since SRCD is specifically designed to improve tail-class generalization, the split-wise performance should be emphasized more systematically, especially on the tail classes. In fact, we have already conducted these split-wise evaluations for the remaining datasets and relevant ablation settings, but did not include all of them in the submission due to space limitations. Here, we provide two representative examples on CIFAR100-LT, where a meaningful tail split exists under the common criterion of classes with fewer than 20 training samples:
>
> | Method    | Head (IR=100) | Med (IR=100) | Tail (IR=100) | All (IR=100) | Head (IR=50) | Med (IR=50) | Tail (IR=50) | All (IR=50) |
> | - | -: |-: |-: |-: |-: |-: |-: |-: |
> | GLMC      |         63.39 |        60.95 |         34.63 |        53.91 |        64.73 |       62.10 |        38.19 |       58.87 |
> | + SRCD    |     **63.88** |    **62.37** |     **36.90** |    **55.26** |    **65.51** |   **63.53** |    **41.86** |   **60.43** |
> | CE+H2T    |         50.85 |        48.91 |         24.53 |        42.27 |        51.42 |   **50.96** |        29.93 |   **47.58** |
> | + SRCD    |     **51.93** |    **51.17** |     **28.16** |    **44.53** |    **51.47** |       49.83 |    **30.21** |       46.97 |
> | BNS  |         62.40 |        59.70 |         31.90 |        52.40 |    **61.30** |       58.70 |        37.20 |       55.90 |
> | + SRCD    |     **62.93** |    **60.56** |     **35.52** |    **53.88** |        61.20 |   **58.94** |    **38.17** |   **56.12** |
> | Focal-SAM |         63.90 |        53.00 |         32.50 |        50.70 |        60.20 |       58.00 |        36.20 |       54.50 |
> | + SRCD    |     **64.66** |    **54.27** |     **36.40** |    **52.54** |    **60.24** |   **58.32** |    **37.15** |   **55.29** |
>
> We show these two settings here because CIFAR100-LT with IR=100 and IR=50 contain a meaningful tail split, making them the most representative cases for directly evaluating the tail-class behavior of SRCD. In contrast, CIFAR10-LT with IR=100/50/10 and CIFAR100-LT with IR=10 do not contain classes with fewer than 20 training samples, so they do not form a meaningful tail split under this criterion. Nevertheless, we have also computed the split-wise performance for those settings, and we will include all of them in the appendix for completeness.
>
> Overall, these additional results further support that the largest gains are consistently observed on the tail split, which is well aligned with the motivation of SRCD. In the revision, we will supplement the appendix with the complete split-wise comparisons for the remaining datasets, as well as the corresponding ablation results, to make the tail improvements more explicit and comprehensive.
>
> ## 3. On writing clarity
>
> Thank you for the careful reading. We will revise the paper to make the Head/Med/Tail definitions explicit and consistent throughout, where Head, Med, and Tail denote classes with >100, 20–100, and <20 training samples, respectively. We will also correct the typo “Table 5(b)” to “Figure 5(b)”.
>
> Overall, we appreciate these suggestions and will incorporate them in the revised version.

---

> > ### Author Rebuttal · Reviewer_XbCh · 2026-04-03
> >
> > For a comprehensive confirmation, I would appreciate a complete result on Places -LT.
> >
> > Furthermore, for some reason, I cannot see your response to differentiate your work to the following existing work mentioned by the other reviewer. Can you elaborate?
> >
> > [1] Cui J, Liu S, Tian Z, et al. Reslt: Residual learning for long-tailed recognition[J]. IEEE transactions on pattern analysis and machine intelligence, 2022, 45(3): 3695-3706.

---

> > > ### Author Response · Authors · 2026-04-03
> > >
> > > Thank you for these thoughtful questions. We clarify both points below:
> > >
> > > ## About the complete result on Places-LT.
> > >
> > > Thank you for the helpful suggestion. We agree that a more complete evaluation on Places-LT would further strengthen the paper. During the rebuttal period, we were only able to finish part of the additional Places-LT experiments due to the limited time. In particular, PECEL was not included at that stage because it required extra reproduction and tuning on Places-LT, and the training-from-scratch results were also not yet complete. We have now completed both parts. For the training-from-scratch setting, we report results with GCL and H2T, since among the training-from-scratch methods considered in our paper, these are the only two methods that have been evaluated on Places-LT; to make the comparison more complete, we also additionally re-implemented BCL and GLMC on Places-LT. For the fine-tuning setting, we now report PECEL. We present the current results below, and we will include a more complete set of Places-LT experiments in the revised version.
> > >
> > > | Method                       | Head      | Med       | Tail      | All       |
> > > | ---------------------------- | --------- | --------- | --------- | --------- |
> > > | Training from scratch        |           |           |           |           |
> > > | BCL                          | **41.98** | 42.36     | 34.91     | 40.77     |
> > > | + SRCD                       | 41.85     | **42.79** | **36.23** | **41.18** |
> > > | GLMC                         | 42.09     | 42.88     | 35.27     | 41.12     |
> > > | + SRCD                       | **42.28** | **43.05** | **37.11** | **41.62** |
> > > | GCL                          | 38.64     | 42.59     | 38.44     | 40.30     |
> > > | + SRCD                       | **38.93** | **42.74** | **39.02** | **40.65** |
> > > | H2T                          | 41.96     | **42.87** | 35.33     | 40.95     |
> > > | + SRCD                       | **42.20** | 42.64     | **38.29** | **41.64** |
> > > | Fine-tuning foundation model |           |           |           |           |
> > > | PECEL                        | 50.1      | 51.2      | 48.3      | 50.2      |
> > > | + SRCD                       | **50.3**  | **51.4**  | **49.2**  | **50.6**  |
> > >
> > > These results show that SRCD remains effective on Places-LT in both settings, with especially clear gains on the medium and tail splits.
> > >
> > > ## About the relation to the existing work mentioned by the Reviewer Eiig.
> > >
> > > Thank you for raising this point. Reviewer Eiig was added after the rebuttal period, so we were unable to respond directly. We therefore clarify it here. We believe the comparison to ResLT is not appropriate. ResLT is fundamentally a multi-branch classification method, where the main branch makes a base prediction and the additional branches compensate for the medium- and tail-class predictions. Its residual learning is therefore essentially a classifier-level output fusion design. In contrast, our work studies shortcut learning as a distinct failure mode in long-tailed recognition. SRCD operates in the CAM space, using head-class proxy CAMs to build a more reliable teacher and distilling it to the tail-class CAM to mitigate shortcut reliance. Thus, SRCD is essentially an evidence-level method. The two works are fundamentally different in motivation, research focus, and technical design, and therefore should not be viewed as the same type of method.
> > >
> > > We sincerely thank you again for your careful reading and valuable suggestions !

---

### Official Review · Reviewer_Eiig · 2026-04-03

**Soundness:** 2
**Presentation:** 2
**Significance:** 2
**Originality:** 2
**Overall Recommendation:** 3
**Confidence:** 1

**Summary:**

This paper proposes Shortcut-Resistant CAM Distillation, which is a plug-and-play framework that transfers object-focused explanations from head to tail classes. The author also provides a theoretical analysis to connect the proposed shortcut reliance to SRCD. The proposed method is evaluated on benchmark datasets to validate the effectiveness of the proposed method.

**Compliance With Llm Reviewing Policy:**

Affirmed.

**Final Justification:**

After reading the rebuttal and the comments from the other reviewers, I decided to maintain my score and believe the paper is below the bar of ICML.

**Key Questions For Authors:**

See above.

**Limitations:**

yes

**Strengths And Weaknesses:**

Pros.
- The paper is well-organized and clearly written.
- The experiments are extensive and helpful to validate the effectiveness of the model.

Cons.
- The novelty of the proposed technique is limited since there were some prior works [1] that incorporated residual learning with long-tailed recognition.
- The ablation studies are not sufficient. More results on more backbones and datasets should be included in Table 3.
- The performance seems to decrease in some cases. More explanation should be included in the paper.

[1] Cui J, Liu S, Tian Z, et al. Reslt: Residual learning for long-tailed recognition[J]. IEEE transactions on pattern analysis and machine intelligence, 2022, 45(3): 3695-3706.

---

### Decision · Program_Chairs · 2026-04-30

**Decision:**

Accept (regular)

**Comment:**

The paper addresses the long-tailed recognition problem by the lens of shortcut learning, and proposes Shortcut-Resistant CAM Distillation (SRCD) to optimize tail class recognition performance. SRCD first generate CAMs from a set of predicted head classes for a given tail sample, and then aggregates the proxy CAMs as a teacher to suppress shortcut reliance for CAM of the tail class. Experimental results demonstrate consistent improvements when SRCD is added to various long-tailed learning baselines.

Three reviewers were positive towards the paper, while one was negative. However, the negative reviewer assigned a very low confidence to the review (score of 1) and only submitted the review at the end of the rebuttal period, leaving the authors with no time to properly address the concerns. Hence this review was given less weight.

The three positive reviewers like the idea of combining long-tailed recognition and CAMs, the theoretical soundness of both the method and the analysis, and the performance of the  method. They raised a number of questions that were successfully answered by the rebuttal.